# Evaluating post-glacial bedrock erosion and surface exposure duration by coupling in-situ OSL and [10]Be dating

Benjamin Lehmann[1], Frédéric Herman[1], Pierre G. Valla[2,3], Georgina E. King[1], and Rabiul H. Biswas[1]

[1]Institute of Earth Surface Dynamics, University of Lausanne, Lausanne, 1012, Switzerland.
[2]University of Grenoble Alpes, Univiversity of Savoie Mont Blanc, CNRS, IRD, IFSTTAR, ISTerre, 38000 Grenoble, France.
[3]Institute of Geological Sciences and Oeschger Centre for Climate Change Research, University of Bern, Bern, 3012, Switzerland.

**Correspondence:** Benjamin Lehmann (lehmann.benj@gmail.com)

**Abstract.** Assessing the impact of Quaternary glaciation at the Earth's surface implies understanding of the long-term evolution of alpine landscapes. In particular, it requires simultaneous quantification of the impact of climate variability on past glacier fluctuations and on bedrock erosion. Here we present a new approach for evaluating post-glacial bedrock surface erosion in mountainous environments by combining terrestrial cosmogenic nuclide [10]Be (TCN) and optically stimulated luminescence (OSL) surface exposure dating. Using a numerical approach, we show how it is possible to simultaneously invert bedrock OSL signals and [10]Be concentrations into quantitative estimates of post-glacial exposure duration and bedrock surface erosion. By exploiting the fact that OSL and TCN data are integrated over different timescales, this approach can be used to estimate how bedrock erosion rates vary spatially and temporally since glacier retreat in an alpine environment.

## 1 Introduction

During the last few million years of the Earth's history, global climate cooled and evolved towards cyclic glaciations in high-latitude and high-altitude regions (e.g., Miller et al., 1987; Zachos et al., 2001; Lisiecki and Raymo, 2005, 2007). It has been suggested that rates of erosion varied during these multiple cycles, and that such variations could in turn feedback into climate (e.g., Molnar and England, 1990; Raymo and Ruddiman, 1992; Champagnac et al., 2007; Herman and Champagnac, 2016). Such erosion rate variations are most expressed in alpine environments, where the main erosion agents vary from ice to water and landslides, during glacial and interglacial periods respectively. However, quantifying how their respective contributions in sediment production have varied remains challenging because both ice-extent fluctuations and associated bedrock surface erosion must be reconstructed simultaneously.

Glacially-polished bedrock offers the possibility to reconstruct past ice-extents and quantify concomitant bedrock surface erosion. These landforms are smooth and glossy, resulting from glacial abrasion, quarrying and melt-water erosion during glacial periods (e.g., Bennett and Glasser, 2009; Siman-Tov, 2017). Following ice retreat, they are exposed to post-glacial erosion, which results in the transition from a well-preserved glacially-polished surface (Figs. 1a and 1b) to a coarse-grained rough surface (Figs. 1c and 1d). Post-glacial bedrock surface erosion is due to the alteration of rock surfaces exposed to atmospheric conditions. Rock alteration can occur through different ways, involving physical (e.g., frost-cracking), chemical and biologi-

cal processes that weaken and modify the rock surface and ultimately results in its progressive erosion (e.g., Łoziński, 1909; Anderson and Anderson, 2010; Hall et al., 2012; Moses et al., 2014). Because we are concerned with the removal of bedrock surface material since exposure to the atmosphere following glacial retreat, rather than the modification of its physical and chemical characteristics caused by weathering, we hereafter use the term "erosion". Our objective is to develop an approach

that may be used to address the following questions: How fast is the transition from a polished bedrock to a coarse-grained surface (Fig. 1)? How much information about postglacial exposure is preserved on weathered rock surfaces? What analytical tools or approach can we use to quantify this morphological transition?

Analytical methods to quantify erosion of rock surfaces differ depending on the timescale of interest (see Moses et al., 2014 for a complete review). Over short timescales (from a few seconds to decades) erosion can be quantified through remote sensing

(e.g., photogrammetric methods; Terrestrial Laser Scanner; c.f., Armesto-Gonzàlez et al., 2010; Duffy et al., 2013) or measured relatively to anthropogenic reference features (historic or experimental; e.g., Nicholson, 2008; Häuselmann, 2008; Stephenson and Finlayson, 2009). Over longer timescales ($10^3$-$10^7$ years), erosion can be measured relative to a natural reference feature (e.g., resistant mineral veins such as quartz or a surface of known age), or quantified using surface exposure dating with terrestrial cosmogenic nuclides (TCN; Lal, 1991; Balco et al., 2008; Bierman and Nichols, 2004; Brandmeier et al. 2011; Liu

and Broecker, 2007). TCN methods rely on the production of specific isotopes in terrestrial material by cosmic rays at or near the Earth's surface (Gosse and Philips, 2001), such as minerals located in the top few meters of soil or bedrock (Lal and Peters, 1967). In glacial and paraglacial environments, the formation of glacial landforms can be directly dated over timescales of $10^3$ to $10^6$ years with TCN surface exposure dating (Ivy-Ochs and Briner, 2014). However, TCN concentrations must also be corrected for surface erosion, which would otherwise lead to an underestimation of the exposure age (Gosse and Phillips, 2001).

The combination of short-lived radionuclides such as $^{14}$C with long-lived radionuclides (i.e., $^{10}$Be, $^{26}$Al, $^{36}$Cl) can be used to resolve and quantify complex exposure histories with burial episodes, but this approach does not allow the quantification of erosion during exposure (Hippe, 2017).

Consequently, complementary approaches are still needed to quantify bedrock erosion over multiple timescales, and more specifically methods that can bridge short and long timescales. In this study, we couple TCN with optically stimulated lumines-

cence (OSL) dating. Rock surface exposure dating using optically stimulated luminescence (named hereafter as OSL surface exposure dating) has recently shown promising potential (e.g., Sohbati et al., 2012a; 2018; King et al., 2019). Luminescence dating is based on the accumulation of trapped electrons through time in the crystalline lattice of specific minerals (e.g., quartz or feldspar), which are sensitive to daylight (Aitken, 1985; Huntley et al., 1985). In addition to its common application to date sediment burial in a range of geomorphological environments (e.g., Duller, 2008; Rhodes, 2011; Fuchs and Owen, 2008),

luminescence dating can also be used to determine the exposure of both naturally formed and anthropogenically formed rock surfaces (e.g., Polikreti et al., 2003; Sohbati et al., 2011; Gliganic et al., 2018; Lehmann et al., 2018). This latter application is based on the principle that when a rock surface is exposed to daylight, the luminescence signal, which is initially homogenous within the rock, will progressively decrease at depth until completely zeroed, a phenomenon called "bleaching" (Aitken, 1998). The assumption is that the longer a surface has been exposed to daylight, the deeper the OSL signal bleaching will be (Polikreti

et al., 2002). In granitic and gneissic rocks, bleaching through time has been shown to occur over the first few millimeters to

centimeters below the rock surface (Vafiadou et al., 2007; Sohbati et al., 2011; Freiesleben et al., 2015). Due to attenuation of daylight, the bleaching rate decreases exponentially with depth. It becomes negligible at depth where the luminescence signal is effectively unbleached and remains in field saturation. For long timescales, trapping due to ionizing radiation will compete with detrapping due to daylight exposure at all depths (after $\sim 10^4$ $a$ in Fig. A1), ultimately resulting in an equilibrium bleaching profile (after $\sim 10^6$ $a$ in Fig. A1, cf. Sohbati et al., 2012a).

For a bedrock OSL profile which is not in equilibrium, measuring and calibrating the depth-dependent luminescence signal beneath the exposed surface by generating multiple luminescence depth profiles enables estimation of an apparent exposure age. OSL surface exposure dating is thus presented as a relatively new surface exposure dating method and has already been applied on both geological and archaeological rock surfaces (Polikreti, 2007; Sohbati et al., 2012a; Freiesleben et al., 2015; Lehmann et al., 2018; Meyer et al., 2018; Gliganic et al., 2018). Sohbati et al. (2012c) were able to quantify the exposure age of historic rock art from the Great Gallery rock art panel in Canyonlands National Park (southeastern Utah, USA). Some of the paintings were damaged by a rockfall event, and conventional luminescence was applied on a rockfall boulder and buried sediments (Chapot et al., 2012). This provided a minimum age for the event. Using a road cut of known age to constrain the bleaching rate for this specific site and lithology, Sohbati et al. (2012c) were able to quantify the exposure age of both the modern analogue ($\sim$130 $a$) and the rock art ($\sim$ 700 $a$). In a periglacial environment, Lehmann et al. (2018) showed that the infrared stimulated luminescence at 50°C (IRSL50) signals from crystalline bedrock slices exhibit increasingly deep bleaching profiles with elevation and thus exposure age, which is consistent with progressive glacier thinning since the Little Ice Age (LIA, $10^1$-$10^2$ $a$). Note that several signals can be targeted in the same rock slice depending on the mineral (e.g., Sohbati et al., 2015; Jenkins et al., 2018). OSL is usually used to analysed the luminescence of quartz (Murray and Wintle, 2000) and IRSL for potassium-rich feldspar signal (both at 50°C and 225°C, Buylart et al., 2009).

Recently, Sohbati et al. (2018) showed that surface erosion has to be taken into consideration when OSL surface exposure dating is applied to natural bedrock surfaces. Indeed, removal of material would bring the bleaching front towards the surface, which may lead to a considerable underestimation of the OSL surface exposure age if not accounted for. When bedrock surface erosion is high ($> 10^{-2}$ $mm$ $a^{-1}$), the competition between bleaching and surface removal will potentially prevent the use of OSL surface exposure dating as a chronometer for bedrock surface exposure (Sohbati et al., 2018). In practice, when erosion is maintained long enough, an equilibrium between trapping, bleaching (i.e., detrapping) and erosion is reached, consequently the bleaching profile reaches steady state. Sohbati et al. (2018) explained that the sensitivity difference to erosion between TCN and OSL surface exposure dating can be exploited to calculate erosion rate experienced by rock surfaces. Indeed, TCN dating is mainly sensitive to cosmic rays over the top $\sim$50-60 $cm$ below the exposed bedrock surface (depending on rock density; Lal et al., 1991) while OSL surface exposure dating is sensitive to light penetration of only millimeters to centimeters (Sohbati et al., 2011, 2012a, 2012b). Thus, using both OSL surface exposure dating and TCN methods, it is possible in theory to quantify surface erosion over different timescales (i.e., $10^2$ -$10^4$ $a$).

Here we couple TCN and OSL surface exposure dating to quantify post-glacial erosion in paraglacial environments. To achieve this, we developed a new model which depends on the exposure age, the surface erosion, the trapping and detrapping (bleaching) rates and the athermal loss (c.f., Eq. 1, Section 2.1). Using this model, we then investigate different synthetic

scenarios in which erosion rates follow a series of step functions in time. After this synthetic experiment, the model is used to invert OSL surface exposure data from two glacially polished bedrock surfaces sampled along the Mer de Glace glacier (Mont Blanc massif, European Alps). We find that the relationships between the depth of luminescence bleaching, the exposure age and the surface erosion allow discrimination between transient and steady state regimes. Finally, we discuss our findings regarding post-glacial surface erosion in paraglacial environments, and the benefits of OSL surface exposure dating combined with TCN surface exposure dating.

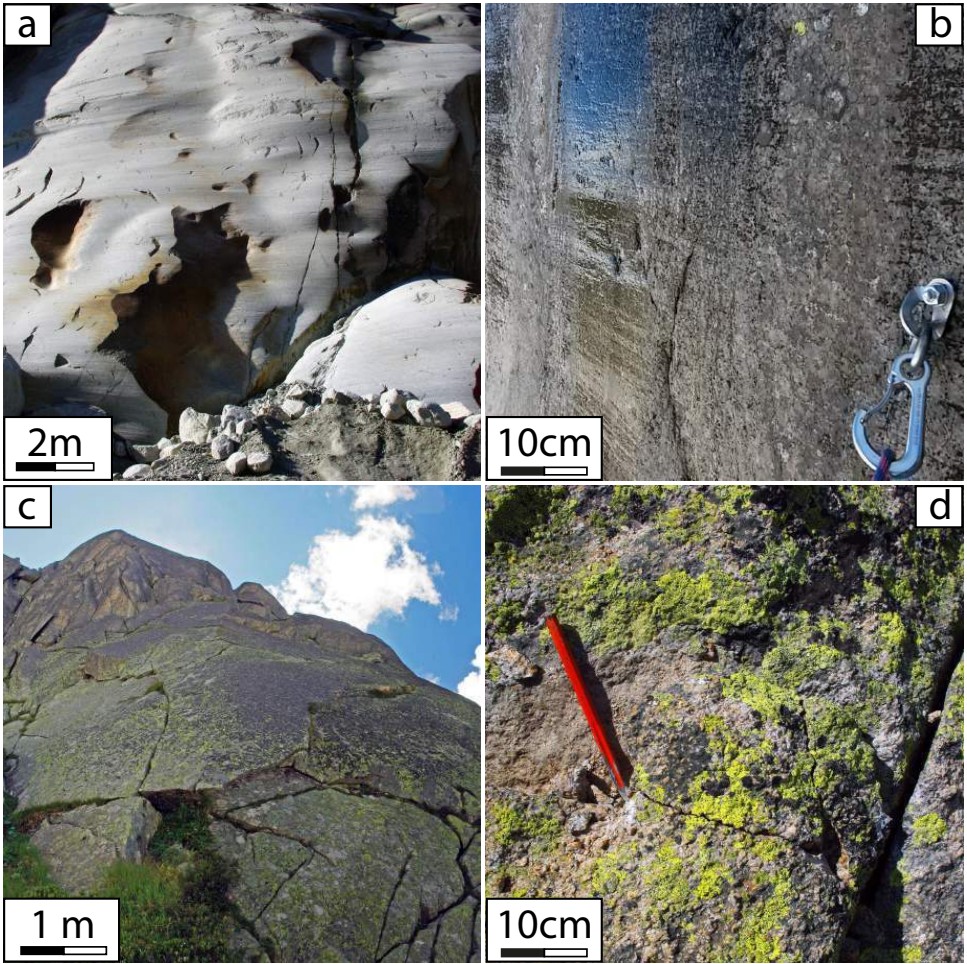

**Figure 1.** Granitic bedrock surfaces along the Mer de Glace glacier (Mont-Blanc massif, European Alps). Surfaces (a) and (b) present well-preserved glacial morphologies exposed for only a few years (striations). Surfaces with longer aerial exposure (Late Glacial to Holocene timescales) show glacially abraded surfaces at the macro-scale (c), but at the $cm$-scale they reveal a coarse-grain rough surface (d).

## 2 Methodology: combining TCN and OSL surface exposure dating

In the following, we focus on the theoretical aspects of both OSL and TCN surface exposure dating methods. We show how different time-dependent exposure and erosion histories are recorded by each technique. Finally, we combine OSL surface exposure and TCN dating to constrain erosion rate and exposure duration simultaneously. Note that all the symbols used below are defined in Table 1.

### 2.1 OSL surface exposure dating

#### 2.1.1 The bleaching model

The intensity of a luminescence signal reflects the number of trapped electrons (Aitken, 1985). For a rock surface exposed to daylight, the luminescence signal intensity, i.e., the trapped electron concentration, is controlled by the competing processes of electron trapping in response to ambient radiation and electron detrapping due to daylight exposure combined with anomalous fading for feldspar IRSL (Habermann et al., 2000; Polikreti et al., 2003; Sohbati et al., 2011). Sohbati et al. (2011, 2012a, b) introduced a mathematical model that describes the process of luminescence bleaching with depth in a homogeneous lithology, enabling the quantification of rock surface exposure duration. Here we propose a new model describing the evolution of luminescence in rock surface as a function of different parameters characterizing the probability of charge trapping, the wavelength-specific photon flux ($\varphi$), the mineral- and wavelength-specific photo-ionization cross-section ($\sigma$) and the lithology-specific light attenuation factor ($\mu$) (Eq. 1). Thus, the measured luminescence signal $L(x,t,r')$ [dimensionless] at given depth $x$ [$mm$], time $t$ [$a$] and recombination center distance $r'$ [dimensionless], can be described by the following differential equation:

$$\frac{dL(x,t,r')}{dt} = \frac{\dot{D}}{D_0}[1 - L(x,t,r')] - L(x,t,r')\overline{\sigma\varphi_0}e^{-\mu x} - L(x,t,r')se^{-\rho'^{-\frac{1}{3}}r'} + \dot{\varepsilon}(t)\frac{dL(x,t,r')}{dx} \tag{1}$$

The first term on the right-hand side of Eq. (1) describes the electron-trapping rate in response to ambient radiation with $\dot{D}$ ($x$) the environmental dose rate [$Gy\ a^{-1}$] at depth $x$ [$m$] and $D_0$ the characteristic dose [$Gy$]. In the context of bedrock surface exposure dating, the dose rate can be approximated as a depth-independent constant in the case of homogeneous lithology i.e., $\dot{D}(x) = const$ (e.g., Sohbati et al., 2018).

The second term describes the electron-detrapping or bleaching rate due to daylight exposure where $\sigma(\lambda)$ is the luminescence photoionization cross section [$mm^2$] defining the probability of a specific trap being excited by light stimulation. $\varphi_0\ (\lambda, x)$ is the photon flux [$mm^{-2}\ a^{-1}$] as a function of wavelength at the rock surface ($x = 0$) and describes the rate of incoming photons that can bleach the trap of interest. Here we assume that the photon flux does not fluctuate through time (Sohbati et al., 2011). We are only concerned with $\overline{\sigma\varphi_0}$ [$a^{-1}$], which is the effective decay rate of luminescence at the rock surface following exposure to a particular light spectrum (Sohbati et al., 2011). The light attenuation coefficient $\mu$ [$mm^{-1}$] describes how deep into the rock a photon will penetrate and affect the luminescence signal. $\mu$ is assumed to be independent of wavelength in the spectral range of interest (Sohbati et al., 2011).

The third term on the right-hand side of Eq. (1) represents the athermal loss of the IRSL signal of feldspar thought to be due to quantum mechanical tunneling of trapped electrons (Wintle, 1973; Visocekas et al., 1998) to the nearest available recombination centers (Huntley, 2006). $s$ is the frequency factor equal to $3 \times 10^{15}$ $s^{-1}$, and $\rho'$ is the dimensionless recombination center density (Tachiya and Mozumder, 1974; Huntley, 2006).

The fourth term describes the advection of the luminescence signal in response to erosion $\dot{\varepsilon}(t) = dx/dt$ $[mm\ a^{-1}]$ on the propagation of the luminescence bleaching front into the rock, using a Eulerian system of reference. Equation (1) is solved using the finite difference method including a second-order upwind scheme for the advection term. This approach is different to the one recently proposed by Sohbati et al. (2018), who used an analytical solution that is based on a confluent hypergeometric function and that requires steady erosion rates. We benchmarked our approach against that of Sohbati et al. (2018) and we

obtain results which are similar to their results calculated using their an analytical solution (Fig. A3).

Ou et al. (2018) experimentally derived $\mu$ for different rock types (greywacke, sandstone, granite and quartzite) using both direct measurements with a spectrometer and bleaching experiments. They showed that the attenuation coefficients are different according the energy of stimulation (e.g., IRSL measured at 50°C and the post-IR IRSL signal measured at 225°C). Meyer et al. (2018) and Gliganic et al. (2018) have shown that the distribution of opaque minerals between rock slices can

significantly affect the reproducibility of luminescence-depth profiles. They conclude the need for close petrographic analysis of luminescence-depth profile samples to ensure that the rock cores from calibration and application sites have a similar mineralogical composition and therefore share similar $\mu$ parameter. In this study, we refer to Sohbati et al. (2011, 2012a) for a complete description of $\overline{\sigma\varphi_0}$ and $\mu$ parameters and their control on the penetration of the bleaching front into a rock surface.

Alternatively, $\overline{\sigma\varphi_0}$ and $\mu$ can be determined from a known-age rock surface with no erosion ($\dot{\varepsilon}(t) = 0$) with a uniform

lithology (Sohbati et al., 2012a; Lehmann et al., 2018; Meyer et al., 2018) and a negligible contribution of athermal loss (as presented in Fig. A2). Under these conditions, Sohbati et al. (2012a) proposed the following analytical solution for Eq. (1), neglecting the athermal loss:

$$L(x,t) = \frac{\overline{\sigma\varphi_0}e^{-\mu x}e^{-t(\overline{\sigma\varphi_0}e^{-\mu x} + \frac{\dot{D}}{D_0})} + \frac{\dot{D}}{D_0}}{\overline{\sigma\varphi_0}e^{-\mu x} + \frac{\dot{D}}{D_0}} \qquad (2)$$

For non-eroding surfaces, OSL surface exposure dating can theoretically be used for a broad range of timescales from 0.01

to $10^5$ years (Fig. A1, and Sohbati et al., 2012a, 2012b, 2018). Under these geomorphic conditions for natural rock surfaces (e.g., glacially-polished bedrock), OSL surface exposure dating has been successfully applied by solving Eq. (2) over $10^1$-$10^2$ $a$ timescales (Lehmann et al., 2018; Gliganic et al., 2018). At longer timescales and/or for rock surfaces affected by erosion, the measured OSL signals do not only reflect the exposure age.

### 2.1.2 Sensitivity analysis to model parameters

In this section, we investigate the respective contribution of the different terms in Eq. (1) for the interpretation of a measured OSL bleaching profile. We investigate the sensitivity of the model to athermal loss, trapping rate and erosion. We use $\overline{\sigma\varphi_0} = 129\ a^{-1}$ and $\mu = 0.596\ mm^{-1}$ that were determined from two calibration rock surfaces of similar granitic lithology from the

Mont Blanc massif, with no erosion and known exposure age (Fig. A2). The values $\dot{D} = 8\ Gy\ ka^{-1}$ (Table 2) and $D_0 = 500$ $Gy$ were selected as they are comparable to the average values for samples used in this study.

### 2.1.2.1. Athermal loss

In this section, we investigate the role of athermal loss when constant erosion rates are low (i.e., $10^{-5}\ mm\ a^{-1}$) and high (i.e., $10^1\ mm\ a^{-1}$). In Eq. (1), $\rho'$ is varied between $10^{-10}$ and $10^{-5}$ (natural values vary between $10^{-6.5}$ and $10^{-5.1}$; Valla et al., 2016; King et al., 2018), and is integrated over dimensionless distances, $r'$, ranging from 0 to 2.5 (Kars et al., 2008) in all cases. Four model runs were done to test whether the shape of the bleaching profile (i.e., luminescence signal vs. depth) changes with different athermal loss rates, rather than the absolute luminescence signal intensity level which reduces as $\rho'$ increases. To remove this effect, the luminescence signals were normalized using the steady state luminescence plateau as unity (NLS for Normalized Luminescence Signal; Figs. 2 and A2a). Figure 2 shows that the shape of the IRSL profiles would be indistinguishable within uncertainties for the two end-member athermal fading rates. We thus find that athermal loss is negligible, and it is not included in the following calculations or considered further.

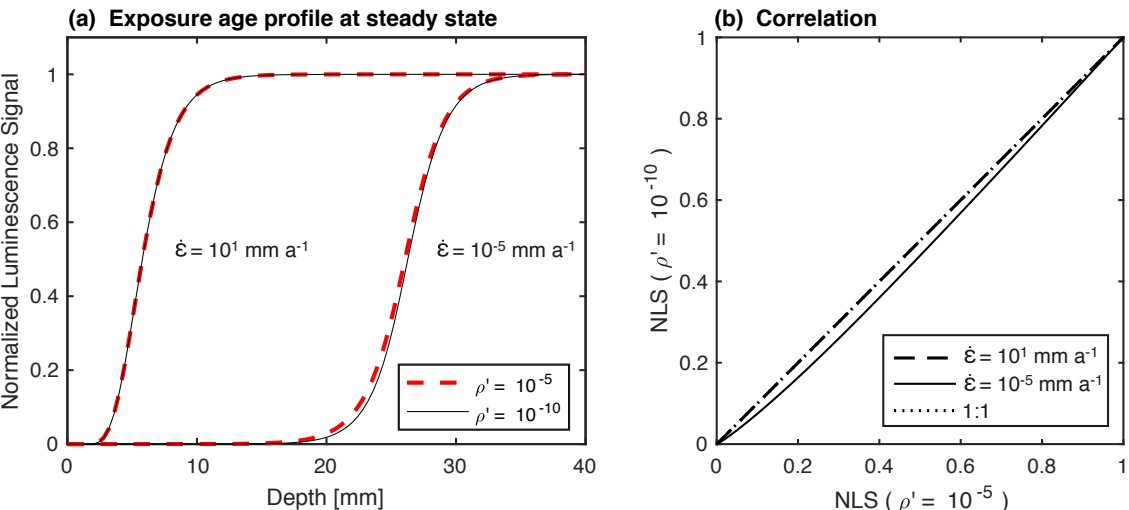

**Figure 2.** (a) Synthetic luminescence profiles predicted by Eq. (1) for two dimensionless recombination center densities $\rho'$ ($10^{-10}$ and $10^{-5}$) and two erosion rates $\dot{\varepsilon}$ ($10^{-10}$ and $10^{-5}\ mm\ a^{-1}$). (b) Comparison of the normalized luminescence signal (NLS) for the different values of $\rho'$ and $\dot{\varepsilon}$. Values for the different parameters $\overline{\sigma\varphi_0}$, $\mu$, $\dot{D}$ and $D_0$ are described in Sect. 2.1.2.

 **2.1.2.2. Trapping**

Here we illustrate the importance of the trapping term and the effect of the different trapping parameters, i.e., the environmental dose rate ($\dot{D}$) and the characteristic dose of saturation ($D_0$), on OSL surface exposure dating. Assuming a non-eroding

rock surface, the bleaching front will keep propagating with time if trapping is not accounted for (Fig. A1; of Sohbati et al., 2012). In contrast, a secular equilibrium (Sohbati et al., 2018) defined by the steady state between trapping and light-stimulated detrapping at depth, can be reached when trapping is considered. In this case, the depth and the time at which the secular equilibrium occurs depends only on $\dot{D}$, $D_0$, $\overline{\sigma\varphi_0}$ and $\mu$ parameters. Using the parameters mentioned in Sect. 2.1.2., and solving Eq.

5    (1) without considering athermal loss, our simulations show that for typical granitic rocks (i.e., $\dot{D}$ between 2 and 8 $Gy\ ka^{-1}$) the bleaching front stabilizes at around 20-25 $mm$ depth after an exposure duration of $10^5$-$10^6$ $a$ (Fig. 3).

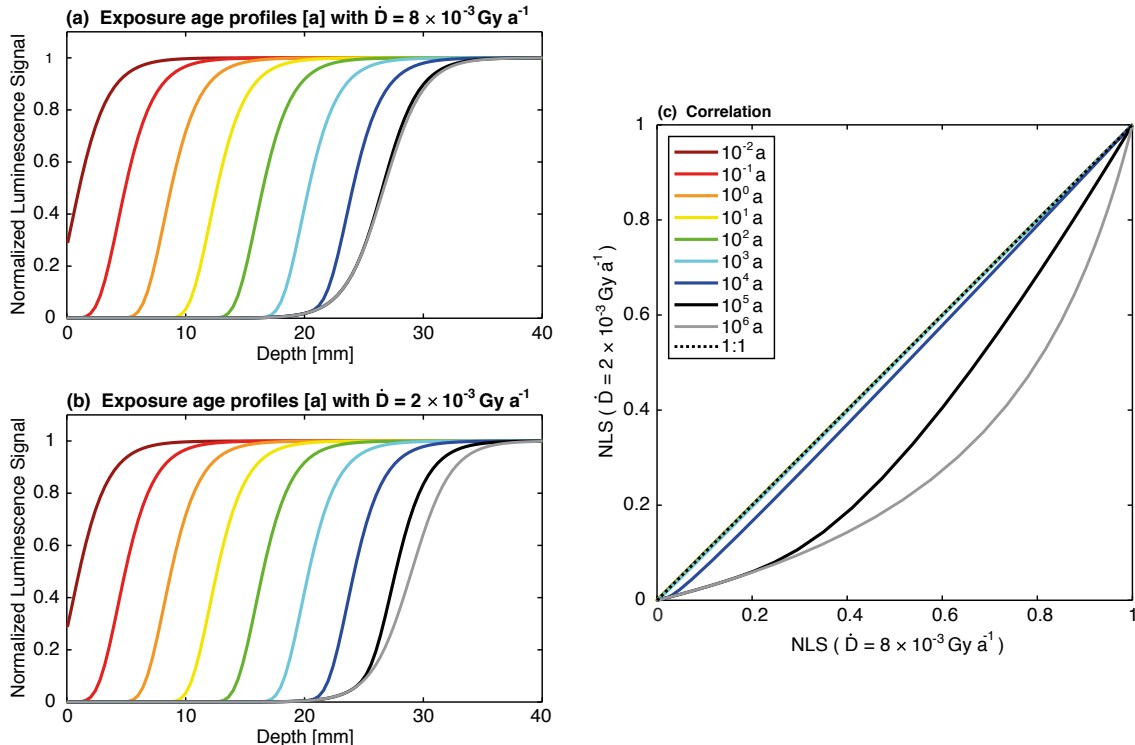

**Figure 3.** Synthetic luminescence profiles for bleaching models with exposure ages from $10^{-2}$ to $10^{-6}$ $a$ and considering trapping rates of (a) $8\times10^{-3}$ and (b) $2\times10^{-3}$ $Gy\ a^{-1}$. Panel (c) shows the comparison of the normalized luminescence signal (NLS) for both models after the different exposure ages. As there is no difference between the modeled profiles for both scenarios between $10^{-2}$ and $10^{-3}$ $a$, the curves are overlying. The choice of parameters $\overline{\sigma\varphi_0}$, $\mu$, $\dot{D}$ and $D_0$ is described in Sect. 2.1.2.

    In Figure 4 we investigate the effects of $\dot{D}/D_0$ on setting the depth of the bleaching front. We use extreme values of $D_0$ of 100 and 2000 $Gy$ and $\dot{D}$ of $2\times10^{-3}$ and $10^{-2}$ $Gy\ a^{-1}$ (King et al., 2016; Jenkins et al., 2018; Biswas et al., 2018), resulting in $\dot{D}/D_0$ from $10^{-6}$ $a^{-1}$ to $10^{-4}$ $a^{-1}$. Our simulations show that the higher the $\dot{D}/D_0$, the closer to the surface the steady-

10   state bleaching profile is which is a consequence of more rapid saturation of the sample luminescence signal. The steady state bleaching depth varies between around 22 and 31 $mm$ (measured at the inflection point) for our end-member simulations (Fig. 4). The influence of $\dot{D}/D_0$ on the bleaching profile is minor relative to the other parameters ($\mu$, $\dot{\varepsilon}$), however, dose rate can vary

by an order of magnitude between rock slices and may possibly explain part of the noise observed in reported experimental data (Meyer et al., 2018).

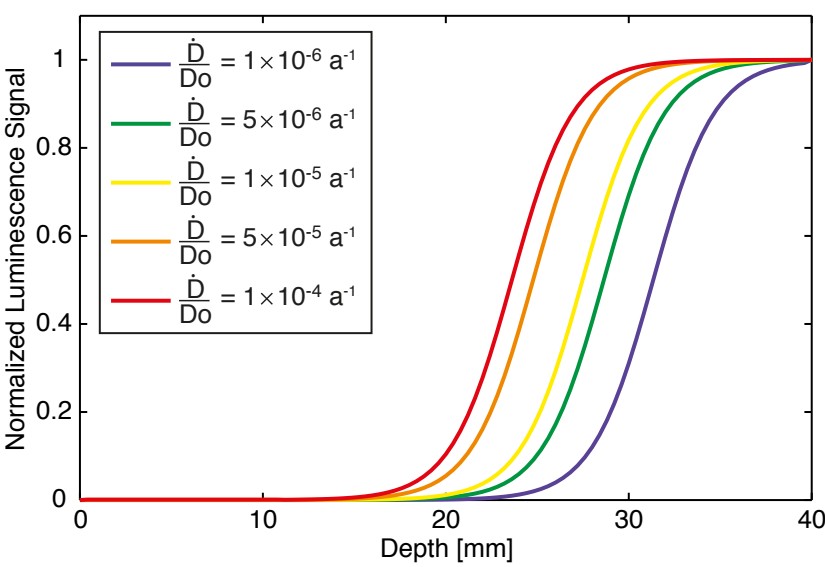

**Figure 4.** Synthetic luminescence profiles predicted by Eq. (1) for different values of the ratio $\dot{D}/D_0$ ($10^{-6}$, $5\times10^{-6}$, $10^{-5}$, $5\times10^{-5}$ and $10^{-4}$ $a^{-1}$) and assuming no erosion. The choice of parameters $\overline{\sigma\varphi_0}$, $\mu$, $\dot{D}$ and $D_0$ is described in Sect. 2.1.2.

**2.1.2.3. Erosion**

The effect of surface erosion on the luminescence signal has recently been highlighted by Sohbati et al. (2018) who proposed an analytical solution to account for this process. In this section, we numerically solve Eq. (1), neglecting athermal loss, and test the effect of different erosion rates on luminescence profiles. Figure 5a shows the resulting synthetic luminescence profiles
at steady state with erosion rates from 0 to $10^2$ $mm\ a^{-1}$. Under these synthetic conditions, the effect of surface erosion starts to be noticeable from around $10^{-4}$ $mm\ a^{-1}$; and for an erosion rate of $10^2$ $mm\ a^{-1}$, the steady state bleaching front is brought forward to 2 $mm$ below the exposed surface. Indeed, surface erosion advects the luminescence signal closer to the surface (Fig. 5). As a result, rock luminescence profiles reflect a competition through time between erosion, trapping and detrapping. When the effects of the three processes are in disequilibrium, such as following initial bedrock surface exposure or onset of
surface erosion, a transient state occurs during which the luminescence signal continues to evolve. After prolonged exposure, and assuming constant erosion, the competing effects equilibrate, leading to a steady state where the bleaching profile is no longer propagating into the rock. In Figure 5b, we evaluate the evolution of the luminescence profiles from transient to steady state using a dimensionless parameter calculated from the product of the profile depth at which luminescence reaches 50% of

its saturation value ($x_{50\%}$), defined as the inflection point NLS($x_{50\%} = 0.5$) and the light attenuation coefficient $\mu$ (Sohbati et al., 2018). We see that the higher the erosion rate is, the faster the system reaches steady state. Consequently, to characterize how a surface is affected by erosion through time, an independent temporal framework is needed to determine the duration of rock surface exposure. This can be achieved through combining OSL surface exposure with TCN dating, which is briefly

introduced in the following section.

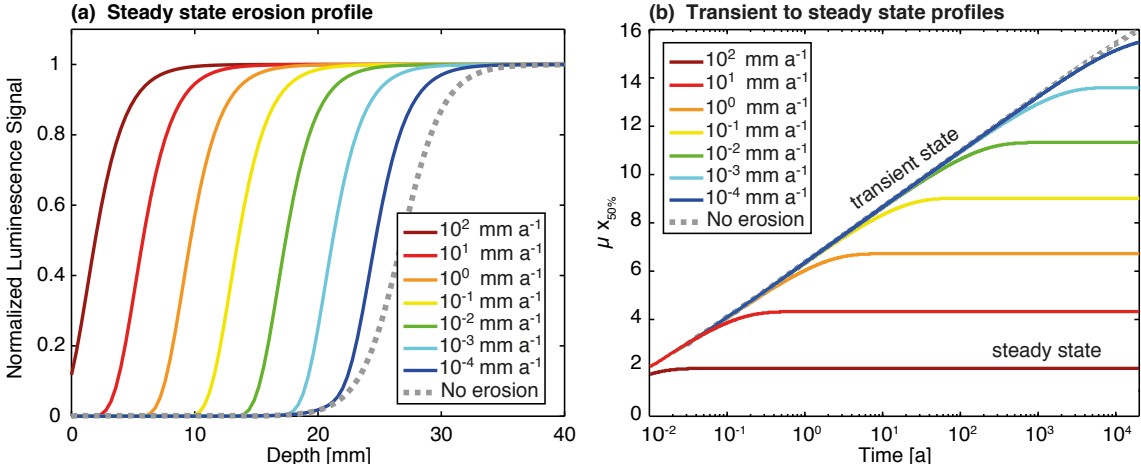

**Figure 5.** Sensitivity of luminescence-depth profiles with erosion. (a) Synthetic luminescence profiles at steady state with erosion rates from 0 to $10^2\ mm\ a^{-1}$. (b) Transient to steady state profile for erosion rates from 0 to $10^2\ mm\ a^{-1}$, as a function of time $[a]$ and as the product of the attenuation factor $\mu\ [mm^{-1}]$ and the depth $x_{50\%}$ defined as NLS($x_{50\%} = 0.5$). The choice of parameters $\overline{\sigma\varphi_0}$, $\mu$, $\dot{D}$ and $D_0$ is described in Sect. 2.1.2.

## 2.2 Terrestrial cosmogenic nuclide (TCN) dating

TCN dating is based on the observation that when cosmic rays reach Earth's surface, they produce cosmogenic isotopes in specific targets, such as the production of $^{10}$Be in quartz (e.g., Gosse and Philips, 2001, Dunai, 2010). The in situ production of quartz $^{10}$Be occurs predominantly within a few meters of Earth's surface and decreases exponentially with depth (Fig. A4a;

Portenga and Bierman, 2011 and references therein). The evolution of cosmogenic nuclide $C$ [atoms $g^{-1}$] in time $t$ $[a]$ and rock depth $x$ $[mm]$ is a function of the disintegration constant $\lambda$ $[a^{-1}]$, the production rate of a radionuclide $P$ [atoms $g^{-1}a^{-1}$] and the erosion $\dot{\varepsilon}$ and can be described by the following equation (Gosse and Phillips, 2001):

$$\frac{dC(x,t)}{dt} = -C(x,t)\lambda + P(0,t)e^{-\nu x} + \dot{\varepsilon}(t)\frac{dC(x,t)}{dx} \tag{3}$$

$P(0)$ is the production rate of the radionuclide at the target surface. The symbol $\nu$ defines the absorption coefficient $[mm^{-1}]$

of the target: $\nu = \frac{\rho}{\Lambda}$. $\Lambda$ is the mean attenuation length for nuclear particles interacting within the target $[g\ mm^{-2}]$. If the

radionuclide concentration at the surface represents the last exposure event, assuming there is no inheritance from a potential previous exposure and that the erosion rate is constant, Eq. (3) can be solved analytically (Lal, 1991), which gives:

$$C(x,t) = \frac{P(0)}{\lambda + \nu\dot{\varepsilon}} e^{-\nu x}[1 - e^{-(\lambda + \nu\dot{\varepsilon})t}] \tag{4}$$

When $t >> 1/(\lambda + \nu\ \dot{\varepsilon})$ the radionuclide concentration reaches a steady state, i.e., a secular equilibrium is reached (Lal, 1991). Under these circumstances, a measured cosmogenic nuclide concentration can be interpreted in terms of a maximum steady-state erosion rate. Here we solve Eq. (3) numerically using the finite difference method, and use the analytical solution to estimate the maximum possible erosion rate. The general behavior of the quartz [10]Be concentration with erosion and exposure age is well documented in the literature (e.g., Lal, 1991), and we illustrate it in Figure A4 for comparison with OSL surface exposure dating (Fig. 5). Note that for solving Eq. (3), the experimental measurement of [10]Be concentration $C_{exp}$ must first be corrected by the depth normalization factor $f_E$ and by the topographic shielding factor $SF$ of the surface following the equation (Martin et al., 2017):

$$C_{corr} = \frac{C_{exp}}{f_E \times SF} \tag{5}$$

with $f_E$ computed by integrating average production over the sample thickness using a single exponential spallation attenuation equation (Balco et al., 2008):

$$f_E = \frac{\Lambda}{\rho \times E}\left[1 - \frac{-\rho \times E}{\Lambda}\right] \tag{6}$$

where $\rho$ is the mean density of the targeted rock [$g\ mm^{-3}$] and $E$ the sample thickness [$mm$]. As we discussed previously, OSL surface exposure and TCN dating both depend on the timing of surface exposure and erosion. These two processes are recorded at different depths into the rock surface: centimeter-scale for OSL surface exposure dating and meter-scale for TCN, therefore OSL surface exposure dating is potentially sensitive to surface erosion over shorter timescales than TCN dating. To combine the two methods, one needs to solve Eqs. (1) and (3) simultaneously, where the two unknowns are the exposure age $t$ and the erosion rate $\dot{\varepsilon}$.

## 3 Inversion approach for synthetic erosion rates

In this section, we generate a series of forward and inverse models. The forward model calculates a luminescence signal and a [10]Be concentration from synthetic erosion and exposure histories. The goal of the inverse model is to constrain the model parameters (i.e., erosion and exposure histories) using the data (i.e., IRSL signal and [10]Be concentration). To validate the inversion procedure, we use the forward model to create synthetic data which we then recover using the inverse model. For these tests, we use the same OSL surface exposure dating parameters explored in the previous sections. $\overline{\sigma\varphi_0} = 129\ a^{-1}$ and $\mu$

**Table 1.** Symbol table

| Symbol | Unit | Description |
|--------|------|-------------|
| **Both methods** | | |
| $x$ | mm | Rock depth |
| $t$ | a | Exposure age |
| $\dot{\varepsilon}$ | mm a$^{-1}$ | Erosion rate |
| $t_S$ | a | Erosion onset time |
| **OSL surface exposure dating** | | |
| $n$ | mm$^{-3}$ | Concentration of trapped charge |
| $L$ | a$^{-1}$ | Maximum possible number of trapped electrons |
| $\sigma$ | mm$^2$ | Luminescence photoionization cross section |
| $\varphi_0$ | mm$^{-2}$ a$^{-1}$ | Photon flux |
| $\mu$ | mm$^{-1}$ | Attenuation coefficient |
| $\lambda$ | mm | Wave of light stimulation |
| $\dot{D}$ | Gy a$^{-1}$ | Environmental dose rate |
| $D_0$ | Gy | Characteristic dose of saturation |
| $s$ | s$^{-1}$ | Frequency factor |
| $\rho^{'}$ | | Dimensionless recombination center density |
| $r^{'}$ | | Dimensionless recombination center distance |
| **TCN dating** | | |
| $t_0$ | a | TCN exposure age without erosion correction |
| $t_C$ | a | TCN exposure age with erosion correction |
| $C$ | atoms g$^{-1}$ | Number of atoms of the radionuclide within the rock |
| $P$ | atoms g$^{-1}$ a$^{-1}$ | Radionuclide production rate |
| $\upsilon$ | mm$^{-1}$ | Absorption coefficient of the specific target |
| $\rho$ | g mm$^{-3}$ | Mean density of the targeted rock |
| $\Lambda$ | g mm$^{-2}$ | Absorption mean free path for nuclear interacting particles in the target |
| $\lambda$ | a$^{-1}$ | Disintegration constant |
| $E$ | mm | Sample thickness |
| $SF$ | | Topographic shielding factor |

$= 0.596\ mm^{-1}$. The value $\dot{D} = 8\times10^{-3}\ Gy\ a^{-1}$ was selected as average value obtained for samples used in this study ($\dot{D} = 7.4$ and $8.4 \times10^{-3}\ Gy\ a^{-1}$ in Table 2). $D_0 = 500\ Gy$ was selected as representative value for IRSL50 signals from granite. The [10]Be exposure age is estimated using the measured quartz [10]Be concentration of sample MBTP1 collected on a polished granitic bedrock surface at 2545 m.a.s.l. from the Tête de Trélaporte located on the left bank of the Mer de Glace glacier

(Mont-Blanc massif, European Alps). Note that the lithology of this sample is similar to that of the OSL surface exposure dating calibration site from which the model parameters are taken (Fig. A2; Lehmann et al., 2018). The sample was located on a surface presenting a shielding factor 0.963 and has a thickness of 8 $cm$ (Table 2). Its non-corrected $^{10}$Be concentration is equal to $474750 \pm 17530$ $at$ $g_{qtz}^{-1}$ using the sea level high latitude (SLHL) rescaled local production rate of the Chironico landslide: $4.16 \pm 0.10$ $at$ $g_{qtz}^{-1}$ $a^{-1}$ (Claude et al., 2014), corrected for the samples' longitude, latitude and elevation and considering no erosion correction and the ERA40 atmospheric model (Uppala et al. 2005). We use a disintegration constant $\lambda$ of $4.9 \times 10^{-7}$ $a^{-1}$, a mean attenuation length for nuclear interacting particles in the target $\Lambda$ of $1.6 \times 10^3$ $g$ $mm^{-2}$ (Gosse and Phillips, 2001; Nishiizumi et al., 2007). The density of the Mont-Blanc granite is measured at around $2.55 \times 10^{-3}$ $g$ $mm^{-3}$.

## 3.1 Forward modeling experiments

In the first scenario, a series of synthetic luminescence profiles were generated using Eq. (1) in a forward model, together with erosion rates of $\dot{\varepsilon} = 10^{-2}$ $mm$ $a^{-1}$ and $\dot{\varepsilon} = 1$ $mm$ $a^{-1}$. This range of values is based on the results of the numerical experiment reported in Sect. 2.1.2.3. For this scenario, erosion rates are assumed to be constant over the TCN exposure age $ts = t_0$, $ts$ being the onset time of erosion (dashed lines in Figs. 7a-d). A reference luminescence profile is also calculated assuming no erosion, using $t_0$ and Eq. (2) (black dot in Fig. 6b and black lines in Fig. 6c and Figs. 7a-d). In the third scenario, another set of synthetic luminescence profiles were again generated using Eq. (1) in a forward model, but the erosion rate was allowed to vary with time (Fig. 6 and green dots in Figs. 7a-d). The assumption made here, is that the evolution of erosion in time can follow a step function (Figs. 6a and 6b). Our objective is explore the effect of a non-constant erosion rate in time on both the luminescence signal and $^{10}$Be concentration.This is the simplest possible time varying erosion rate history. The erosion is initially equal to zero, i.e., between the corrected exposure age $tc$, and an onset time of erosion $ts$, and increase to a fixed rate between $ts$ and today (Fig. 6a). Note that more sophisticated erosion rate histories could be tested with the same approach, which is beyond the scope of the current study. Figure 6 illustrates the schematic representation of four different erosion scenarios through time (Figs. 6a and 6b) and their resulting luminescence signal (Fig. 6c). Note that the corrected exposure age $tc$ is part of the calculation and is obtained by solving Eq. (3) and using the nuclide concentration and an entire erosion rate history. We report the four model outputs calculated using $ts$ between 1 and 100 $a$, and erosion rates $\dot{\varepsilon}$ between $10^{-2}$ and 1 $mm$ $a^{-1}$ (green dots respectively in Figs. 7a-d). Note that we added 10% of white noise to the predicted OSL surface exposure dating profiles (used for the inversion approach in Sect. 2).

By applying a constant erosion rate of $10^{-2}$ $mm$ $a^{-1}$ to a rock surface exposed since $t_0$ ($16428 \pm 707$ $a$), the luminescence signal is brought 7.8 $mm$ closer to the surface (i.e., 17 $mm$ deep from the surface) compared to the reference signal (luminescence signal exposed since $t_0$ and no affected by erosion; black line in Figs. 7a-d at 24.8 $mm$ deep from the surface). For a constant erosion rate of 1 $mm$ $a^{-1}$, the luminescence signal is brought 15.4 $mm$ closer to the surface (i.e., 9.4 $mm$ deep from the surface) compared to the reference signal (difference between black lines and dash lines measured at NLS = 0.5 in Figs. 7a-d).

If an erosion rate of $10^{-2}$ $mm$ $a^{-1}$ is applied for a duration of 1 $a$ before sampling and integrated over its specific corrected exposure age (since $tc = 16428 \pm 707$ $a$), the luminescence signal is brought 0.4 $mm$ closer to the surface compared to the

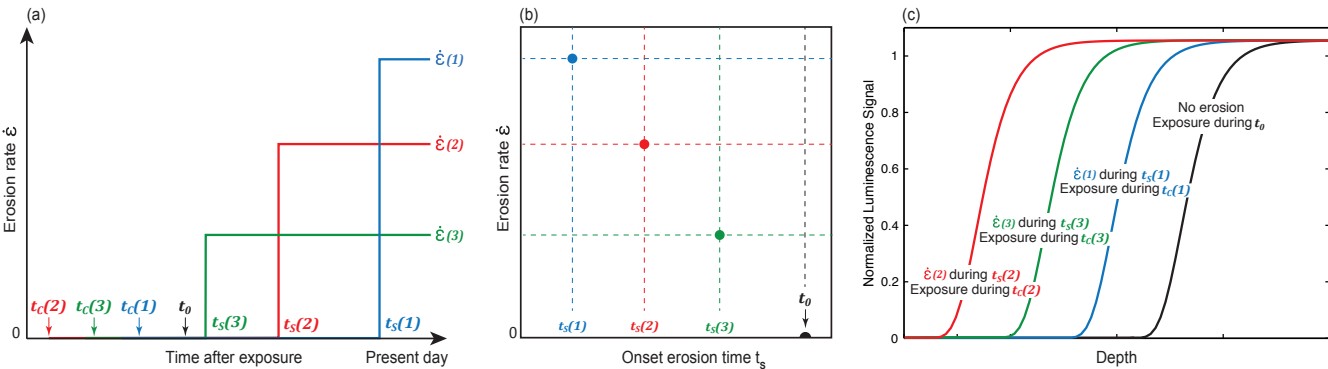

**Figure 6.** Schematic representation of four different erosion scenarios through time (a) and (b) and their resulting luminescence signal (c). $t_0$ is the uncorrected [10]Be exposure age, $ts$ the onset times of erosion, $tc$ the corrected exposure ages, and $\dot{\varepsilon}$ the erosion rate. Note that the luminescence plots in (c) are not model outputs but drawings, with the aim of conceptualizing how the experiments are designed.

reference signal (green dots in Fig. 7a) and 1.2 $mm$ if the same erosion rate is applied for 100 $a$ before sampling and integrated over its specific $tc$ (16455 ± 713 $a$; green dots in Fig. 7b). In both scenarios, the predicted luminescence profiles do not overlap the luminescence profile predicted for a constant erosion rate indicating that the system is in a transient state.

For an erosion rate of 1 $mm\ a^{-1}$ applied during 1 $a$ before sampling and for an exposure time corrected with its specific
erosion history $tc$ (16455 ± 713 $a$), the luminescence profile (green dots in Fig. 7c) is brought 1.2 $mm$ closer to the surface compared to the reference signal (black line in Fig. 7c). In this case, the luminescence profile is in transient state with erosion because it is not overlapping the luminescence profile produced by applying the same erosion rate for an infinite time (dashed line in Fig. 7c). Interestingly, the same effect on the luminescence signal is produced by applying an erosion rate of 1 $mm\ a^{-1}$ during 1 $a$ (green dots in Fig. 7c) and an erosion rate $10^{-2}\ mm\ a^{-1}$ during 100 $a$ before sampling (green dots in Fig. 7b). For
an erosion rate of 1 $mm\ a^{-1}$ applied during 100 $a$ before sampling and for an exposure time corrected with its specific erosion history $tc$ (16945 ± 722 $a$), the luminescence signal is brought 15.4 $mm$ closer to the surface (green dots in Fig. 7d) compared to the reference signal (black lines in Fig. 7d). A similar result is obtained when erosion rate is applied for an infinite time (dashed line in Fig. 7d): in this scenario, the steady state with erosion is reached.

### 3.2   Inverse modeling experiments

The synthetic data are now inverted to assess the extent to which it is possible to recover the values of $\dot{\varepsilon}$ and $ts$. Ultimately, our objective is to establish and validate a numerical protocol that enables erosion rate histories to be estimated from paired OSL surface exposure and TCN dating measurements on bedrock surfaces. To find the most likely solutions, we test $10^4$ pairs of both $\dot{\varepsilon}$ and $ts$ (combination of 100 values of both parameters) in log space. The range of possible erosion rates $\dot{\varepsilon}$ varies between $10^{-5}$ and $10^1\ mm\ a^{-1}$. These end-member values were selected from the erosion sensitivity test performed in Sect. 2.1.5. The

erosion onset times $ts$ range between $5 \times 10^{-1}$ $a$ and $3 \times 10^4$ $a$, this range being arbitrarily decided with the upper boundary set to approximately twice the initial TCN age.

As mentioned above, the measured $^{10}$Be concentration has be to corrected for erosion. If the applied erosion rate is too high or the duration is too long, or both, the $^{10}$Be concentration must remain small (Fig. A4). On that basis, there is a range of solutions with high erosion rates and durations which is unable to predict the observed $^{10}$Be concentration (Lal, 1991). We call this the "forbidden zone", and exclude it from the parameter search. Expressed differently, for each $\dot{\varepsilon}$ and $ts$ pair, Eq. (3) is first solved and a first estimate of the corrected exposure age $tc$ is calculated. However, Eq. (3) does not yield a solution for a range of values that produce too much erosion and thus too high $^{10}$Be concentration loss to fit the measured sample concentration. In the studied cases, the forbidden zone is defined by the values between the pairs of $\dot{\varepsilon} = 10$ $mm$ $a^{-1}$, $ts \sim 110$ $a$ and $\dot{\varepsilon} \sim 5 \times 10^{-1}$ $mm$ $a^{-1}$, $ts = 29210$ $a$.

For all the other pairs of $\dot{\varepsilon}$ and $ts$, the corrected exposure age $tc$ is subsequently used to predict luminescence profiles ($NLS_{inverse}$) that are compared to the synthetic luminescence profiles ($NLS_{forward}$) presented in the previous section (green dots in Figs. 7a-d). The quality of these fits are evaluated using a misfit function and the inversion results are converted into probability density functions using a likelihood function (Eq. 7). The least square deviations regression method minimizes the sum of the square differences between the forward $NLS_{forward}$ and the inverted values $NLS_{inverse}$ giving:

$$\mathcal{L} = exp(-\frac{1}{\sigma^2} \sum_{i=1}^{n} \left[ NLS_{forward}^{(i)} - NLS_{inverse}^{(i)} \right]^2) \qquad (7)$$

Where $n$ is the number of rock slices per sample and $\sigma$ is the standard deviation of the normalized saturated luminescence signal intensities that form the plateau at depth ($0.053 \leqslant \sigma \leqslant 0.059$ for our samples).

The results of these inversions are shown in Figures 7e-h with the parameter space for erosion rate/time and the resulting likelihood. The green circles depict the synthetic forward modelled pair of $\dot{\varepsilon}$ and $ts$ ($NLS_{forward}$) which should be recovered in the inversion (green dots in Fig. 7a-d), and the black circles show the $\dot{\varepsilon}$ - $t_0$ pair used to produce the model assuming erosion is constant (dashed lines in Figs. 7a-d). We then select the pairs of $\dot{\varepsilon}$ and $ts$ leading to the maximum 5% likelihood values which are fitting the synthetic data (the threshold of 5% is arbitrarily chosen), and plot their corresponding luminescence profile values (red lines in Figs. 7a-d).

The first noticeable observation is that the erosion rate $\dot{\varepsilon} = 10^{-2}$ $mm$ $a^{-1}$ could be applied over every time period below $\sim 3 \times 10^{-3}$ $a$. The numerical solutions for both constant and non-constant erosion rate lay outside of the forbidden zone (black and green circles respectively in Figs 7e-f). As another example, an erosion rate equal to $\dot{\varepsilon} = 1$ $mm$ $a^{-1}$ could also be applied for any time lower than 1200 $a$. Indeed, it is not possible to apply an erosion $\dot{\varepsilon} = 1$ $mm$ $a^{-1}$ during $t_0$ as this pair of values would lie in the forbidden zone (Figs 7g, h) since such a high erosion rate would imply too high $^{10}$Be concentration loss to fit the measured sample concentration.

For the first scenario, the synthetic luminescence profile produced by applying an erosion rate $\dot{\varepsilon} = 10^{-2}$ $mm$ $a^{-1}$ during time period $ts = 1$ $a$ has a great number of possible pairs of $\dot{\varepsilon}$ and $ts$ that would reproduce this specific luminescence signal (Normalized likelihood > 0.9: yellow area in Fig. 7e). The acceptable solutions range between pairs of values below $\dot{\varepsilon} \sim 2$

$\times 10^{-2}$ $mm$ $a^{-1}$ with $ts = 5 \times 10^{-1}$ $a$ and $\dot{\varepsilon} = 10^{-5}$ $mm$ $a^{-1}$ with $ts = 10^3$ $a$. These low values do not produce enough erosion to significantly alter the TCN exposure age ($tc \sim t_0$).

In the second scenario, the erosion rate is $\dot{\varepsilon} = 10^{-2}$ $mm$ $a^{-1}$ during a time period $ts = 100$ $a$ and the forward model pair values can be successfully recovered from the inversion with a more restrained range of numerical solutions (Fig. 7f). The transient state with erosion is well illustrated by trade-offs between erosion rate and time. To fit the forward luminescence profile, low erosion rates should be associated with long time periods following the trend from $\dot{\varepsilon} \sim 2$ $mm$ $a^{-1}$ with $ts = 5 \times 10^{-1}$ $a$ to $\dot{\varepsilon} \sim 1.4 \times 10^{-4}$ $mm$ $a^{-1}$ with $ts = 1.2 \times 10^4$ $a$. When the erosion rate of $1.4 \times 10^{-4}$ $mm$ $a^{-1}$ is applied longer than $1.2 \times 10^4$ $a$, a steady state with erosion is reached and this specific erosion rate could be applied for an infinite time. The highest correction of the TCN exposure age possible with these solutions is of the order of 0.1% ($t_0 = 16428 \pm 707$ $a$ and $tc = 16455 \pm 713$ $a$), which is insignificant compared to the 3.6% uncertainties on $t_0$.

The third scenario, where the erosion rate is $\dot{\varepsilon} = 1$ $mm$ $a^{-1}$ during time period $ts = 1$ $a$, shares the exact same solution as the second case ($\dot{\varepsilon} = 10^{-2}$ $mm$ $a^{-1}$ with $ts = 100$ $a$). This confirms the observation made with the forward modeling where both scenarios predicted similar luminescence profile depths. This can be explained because both pairs of $\dot{\varepsilon}$ - $ts$ lie on the trend from $\dot{\varepsilon} \sim 2$ $mm$ $a^{-1}$ with $ts = 5 \times 10^{-1}$ $a$ and $\dot{\varepsilon} \sim 1.4 \times 10^{-4}$ $mm$ $a^{-1}$ with $ts = 1.2 \times 10^4$ $a$.

In the fourth scenario, the erosion rate $\dot{\varepsilon} = 1$ $mm$ $a^{-1}$ is applied during time $ts = 100$ $a$, the range of solutions is much more restrained than for the other scenarios. The synthetic luminescence profile is at steady state with erosion, where the erosion rate $\dot{\varepsilon} = 1$ $mm$ $a^{-1}$ can be applied from 18 to 1200 $a$. For longer time of erosion, the pairs of $\dot{\varepsilon}$ - $ts$ lie within the forbidden zone regarding the TCN concentration. In this case, the maximum correction of the TCN exposure age is around 3.1% ($t_0 = 16428 \pm 707$ $a$ and $tc_{max} = 16945 \pm 722$ $a$), which is comparable to the initial uncertainty on $t_0$.

## 4 Application to natural samples

In this section, we apply the method presented above on two natural rock surfaces. Samples MBTP1 and MBTP6 were collected from glacially-polished bedrock surfaces at 2545 and 2084 m.a.s.l. respectively from the Tête de Trélaporte located on the left bank of the Mer de Glace glacier (Mont-Blanc massif, European Alps). Rock surfaces were collected for application of both the TCN and OSL surface exposure dating methods (Fig. 9 and Tables 2 and 3.3). Both samples are from the same phenocristalline granitic lithology of the Mont Blanc massif (Fig. 8).

### 4.1 Sample preparation, measurement and age calculation

The [10]Be sample preparation method is comprehensively described in the literature (e.g., Kohl and Nishiizumi, 1992; Ivy-Ochs, 1996). We used quartz separates from grain sizes between 250 $\mu m$ and 1 $mm$. The addition of a commercial [9]Be carrier was followed by quartz dissolution in HF and Be purification using ion-exchange columns and selective precipitation. The [10]Be/[9]Be ratio was measured by accelerator mass spectrometry (AMS) on the 600 KV TANDY system at the Laboratory of Ion Beam Physics (LIP) at ETH Zürich (Switzerland) against the standard S2007N (Christl et al., 2013) that is calibrated against the 07KNSTD standard (Nishiizumi et al., 2007). We correct for a long-term average full chemistry procedural blank of

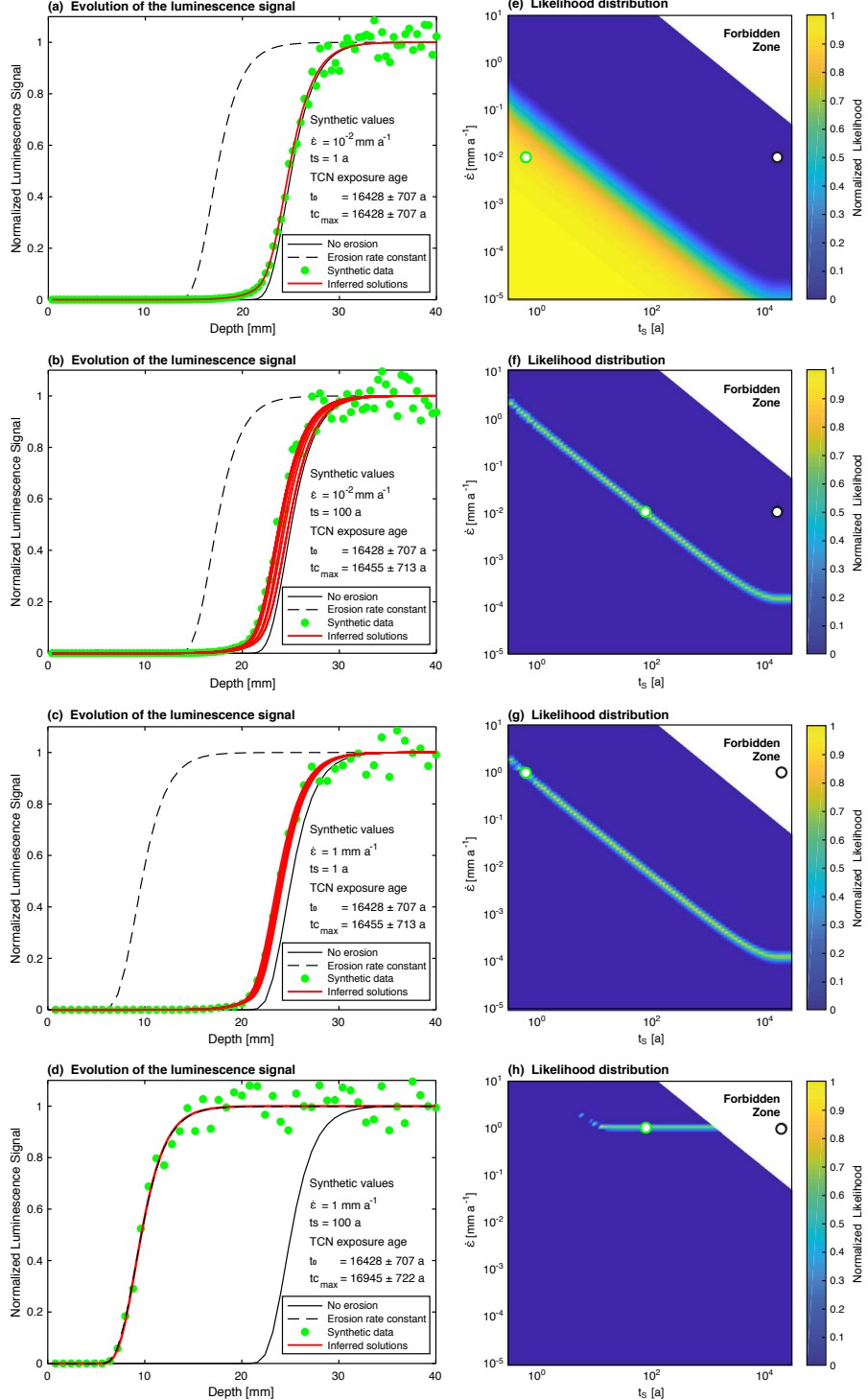

**Figure 7.** Caption on the next page

. **Figure 7 on the previous page:** Results of forward and inverse modeling experiments. Green dots represent the simulated luminescence profiles for rock surfaces exposed to (a) an erosion rate of $\dot{\varepsilon} = 10^{-2}\ mm\ a^{-1}$ during time $ts = 1\ a$, (b) an erosion rate of $\dot{\varepsilon} = 10^{-2}\ mm\ a^{-1}$ during time $ts = 100\ a$, (c) an erosion rate of $\dot{\varepsilon} = 1\ mm\ a^{-1}$ during time $ts = 1\ a$ and (d) an erosion rate of $\dot{\varepsilon} = 1\ mm\ a^{-1}$ during time $ts = 100\ a$. Black lines represent the reference luminescence profiles for a surface exposed since $t_0 = 16428 \pm 707\ a$ with no erosion. Dashed lines show the luminescence profiles produced by applying a constant erosion rates of (a) (b) $\dot{\varepsilon} = 10^{-2}\ mm\ a^{-1}$ and (c) (d) $\dot{\varepsilon} = 1\ mm\ a^{-1}$ during $t_0$. Red lines represent the best-fitting profiles inverted for all numerical solutions with likelihood >5%. $tc_{max}$ represents the maximum corrected TCN exposure age using the forward modeled values of $\dot{\varepsilon}$ and $ts$. (e), (f), (g) and (h) represents the likelihood distributions inverted from the synthetic luminescence profiles respectively in (a), (b), (c) and (d). Green open circles represent the pairs of values of $\dot{\varepsilon}$ and $ts$ used in the forward model to produce the profiles, and the black open circles represent the values $\dot{\varepsilon}$ and $t_0$ used to predict luminescence profiles with constant erosion (dashed lines insets (a), (b), (d) and (c)). All models were performed by solving Eq. (1) using the following parameters: $\overline{\sigma\varphi_0} = 129\ a^{-1}$, $\mu = 0.596\ mm^{-1}$, $\dot{D} = 8{\times}10^{-3}\ Gy\ a^{-1}$ and $D_0 = 500\ Gy$. TCN ages were calculated by solving Eq. (3) for the [10]Be concentration of sample MBTP1 presented in the following section.

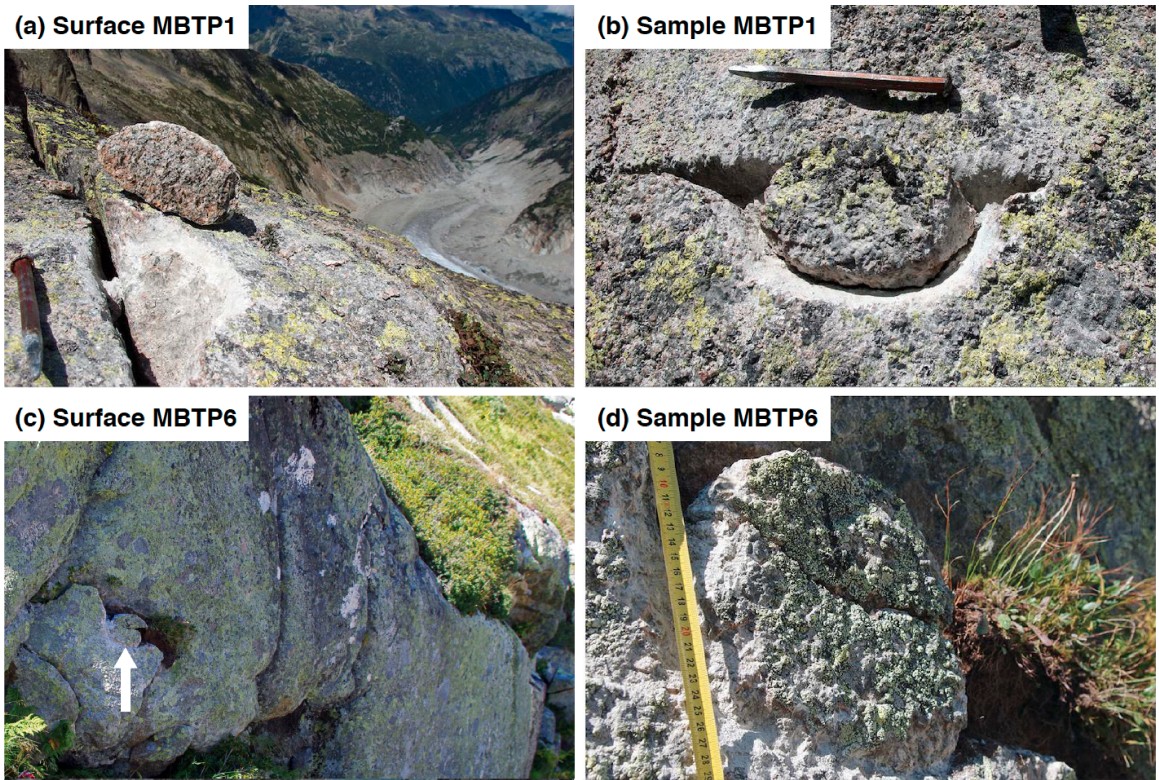

**(a) Surface MBTP1**

**(b) Sample MBTP1**

**(c) Surface MBTP6**

**(d) Sample MBTP6**

. **Figure 8.** Locations and sample pictures of MBTP1 and MBTP6, both located on the Tête de Trélaporte along the Mer de Glace glacier (Mont Blanc massif, European Alps).

[10]Be/[9]Be $(3.7 \pm 2.2) \times 10^{-15}$ . Ages are calculated using the SLHL rescaled local production rate of the Chironico landslide: $4.16 \pm 0.10\ at\ g_{quartz}^{-1}\ a^{-1}$ (Claude et al., 2014), corrected for the samples' longitude, latitude and elevation and considering

no erosion correction, with the Lifton-Sato-Dunai (LSD) scaling scheme (Lifton et al., 2014), the ERA40 atmospheric model (Uppala et al., 2005) and the Lifton VDM 2016 geomagnetic database (for ages between 0-14 $ka$, Pavon-Carrasco et al., 2014 and for ages between 14-75 $ka$, GLOPIS-75, Laj et al., 2004) with a modified version of the CREp online calculator to process non-linear erosion rate correction by solving Eq. (3) (Martin et al., 2017). The reported errors propagate uncertainties from AMS standard reproducibility, counting statistics, the standard mean error of the samples, blank correction and the local production rate. These external errors are used to compare absolute ages to independent chronologies. All errors are reported at $1\sigma$.

For luminescence analysis we followed the methodology of Lehmann et al. (2018). The bedrock samples were cored down to 30 $mm$ depth using a Husqvarna DM220 drill, with 10-$mm$ diameter. Cores were then sliced into 0.7-$mm$ thick rock slices with a BUEHLER IsoMet low speed saw equipped with a 0.3-$mm$ thick diamond blade. The samples were drilled and sliced under wet conditions (water and lubricant, respectively) to avoid any heating that could potentially reset the OSL signal. Sample preparation was done under subdued red-light conditions. The thickness of each rock slice was measured to determine the precise depth of each luminescence measurement. Luminescence measurements were performed using Ris TL-DA 20 TL/OSL readers (Bøtter-Jensen et al., 2010) equipped with [90]Sr beta sources at the University of Lausanne (Switzerland). We performed a preheat at 250°C before giving infrared (IR) stimulation (870 $nm$, FWHM 40 $nm$) at 50°C (the sample preparation and analysis are described in further detail in the Fig. A2 and A5). The calculation of $\dot{D}$ was achieved through the measurement of the concentrations of U, Th, K and Rb of the bulk rock sample and the use of the DRAC online calculator (Table 2 and details in Table A1; Durcan et al., 2015). The determination of $D_0$ was done by constructing a dose response curve (DRC) of the IRSL signal measured at 50°C using a single aliquot regenerative dose (SAR) protocol (Murray and Wintle, 2000; Wallinga et al., 2000) and fitting the DRC with single saturating exponential. The validity of the measurement protocol was confirmed using a dose-recovery experiment (Wallinga et al., 2000). Recovered doses were within 10% of unity.

## 4.2 Experimental results

Sample MBTP1 provided a [10]Be concentration of 474750 $\pm$ 17530 $at\ g_{qtz}^{-1}$. The solution of Eq. (3) gives an apparent [10]Be age for sample MBTP1 of $t_0$ = 16428 $\pm$ 707 $a$ assuming sample thickness of 8 $cm$ and a shielding factor of 0.963 (Tables 2 and 3). In the same way, the measured [10]Be concentration of 84100 $\pm$ 13060 at $g_{qtz}^{-1}$ for sample MBTP6 gives a [10]Be age of $t_0$ = 6667 $\pm$ 965 $a$, assuming a sample thickness of 7 $cm$ and a shielding factor of 0.594 (Tables 2 and 3). Apparent [10]Be ages were calculated as described in Section 4.1, assuming no erosion.

Figure 9 shows the infrared stimulated luminescence at 50°C (IRSL50, Normalized Signal) measurements of samples MBTP1 and MBTP6. Three replicates (i.e., individual cores) per sample were sliced in a way that a depth and an IRSL50 signal can be attributed to each rock slice (Tables A2 and A3). The IRSL50 signal is bleached near the surface and reaches a plateau at depth (even for sample MBTP1 where the plateau is poorly defined). The scattering of the measurements between rock slices is probably due to the granitic nature of the samples. Indeed, the phenocryst lithology can cause heterogeneity in the resulting IRSL50 signals (Meyer et al., 2018) caused by differential bleaching and possibly variations in the environmental dose rate, mainly beta dose heterogeneity (Morthekai et al., 2006) and thus the rate of electron trapping.

**Table 2.** Sample list and measurements

| Sample ID | Latitude | Longitude | Elevation | Thickness | Topographic | $^{10}$Be conc.[a] | $P(0)$ local[b] | $\dot{D}$ spec.[c] |
|---|---|---|---|---|---|---|---|---|
| | WGS 84 | | [m.a.s.l.] | [cm] | Shielding factor | [at $g_{qtz}^{-1}$] | [at $g_{qtz}^{-1}$] | Gy a$^{-1}$ |
| MBTP1 | 45.9083 | 6.9311 | 2545 | 8 | 0.963 | 474750 ± 17530 | 30.20 ± 0.72 | 7.4 10$^{-3}$ |
| MBTP6 | 45.9129 | 6.9326 | 2094 | 7 | 0.594 | 84100 ± 13060 | 21.74 ± 0.52 | 8.4 10$^{-3}$ |

[a] Measured against standard 07KNSTD (Nishiizumi et al., 2007), corrected for full process blank of $(3.7 \pm 2.2) \times 10^{-15}$ $^{10}$Be/$^9$Be.

[b] Local production rate using the sea level high latitude (SLHL) rescaled local production rate of the Chironico landslide: $4.16 \pm 0.10$ at $g_{quartz}^{-1}$ $a^{-1}$(Claude et al., 2014), corrected for the samples' longitude, latitude and elevation and considering no erosion correction, with the LSD scaling scheme (Lifton et al., 2014), the ERA40 atmospheric model (Uppala et al., 2005) and the Lifton VDM 2016 geomagnetic database (for ages between 0-14 $ka$, Pavon-Carrasco et al., 2014 and for ages between 14-75 $ka$, GLOPIS-75, Laj et al., 2004).

[c] Dose rates were calculated using the concentrations of U, Th and K of the bulk rock sample and the DRAC online calculator (details in Table A1; Durcan et al., 2015).

As a reference profile, a model is computed by solving Eq. (2) using $t_0$ and considering no erosion (black line in Fig. 9a) and lies at 25 $mm$ below the rock surface. The bleaching front measured from the IRSL50 signal of sample MBTP1 (green dots in Fig. 9a) is located 4 $mm$ closer to the surface compared to the reference profile (21 $mm$ from the surface). The IRSL50 profile considering no erosion correction gives an apparent age of about 2 orders of magnitude lower compared to $t_0$, about
$642 \pm 160$ $a$ ($1\sigma$; Table 3 and Fig. A5).

For sample MBTP6, the reference profile is at 23.5 $mm$ below the surface (black line in Fig. 9b). The measured IRSL50 profile (green dots in Fig. 9b) is approximately 16.5 $mm$ closer to the surface in comparison to the reference profile (7 $mm$ from the surface). The OSL surface exposure apparent age for sample MBTP6 is about $0.39 \pm 0.02$ $a$ ($1\sigma$; Table 3 and Fig. A5).

**4.3   Inversion results**

In this section, we report the results from the inversion of $\dot{\varepsilon}$ and $ts$ for the IRSL50 profiles of samples MBTP1 and MBTP6 following the procedure presented in Section 2. For both samples, the corrected $^{10}$Be age are calculated using Eq. (3) with a range of erosion rates from $10^{-5}$ and $10^1$ $mm$ $a^{-1}$ and $ts$ ranging from $5 \times 10^{-1}$ $a$ to $10^{log(t_0)+0.25}$ $a$ (this formula limits the search to $\sim 30$ $ka$ because these surfaces are known to be post-LGM; Coutterand and Buoncristiani, 2006).
The resulting forbidden zone for sample MBTP1 lies between the erosion rate/time pairs of $\dot{\varepsilon}$ = 10 $mm$ $a^{-1}$, $ts$ $\sim$110 $a$ and $\dot{\varepsilon}$ $\sim 5 \times 10^{-1}$ $mm$ $a^{-1}$, $ts$ = 29210 $a$ (already discussed in Sect. 3.2). The inversion results indicate that sample MBTP1 reached a steady state with erosion characterized by an erosion rate of $\dot{\varepsilon}$ = $(3.5 \pm 1.2) \times 10^{-3}$ ($1\sigma$) $mm$ $a^{-1}$ applied during a minimum

duration of 2300 $a$ (Fig. 9c). In these conditions, the corrected TCN age is $tc_{ss}$ = 16647 ± 593 $a$ (1.1% of correction). The maximum corrected TCN age $tc_{max}$ = 17396 ± 746 $a$ is obtained by using $\dot{\varepsilon}$ = (3.5 ± 1.2)×10$^{-3}$ (1$\sigma$) $mm\ a^{-1}$ and the maximum $ts$ possible (29214 $a$), this comprises a correction of about 5.8%.

For sample MBTP6, the forbidden zone lies in between the erosion rate/time pairs of $\dot{\varepsilon}$ = 10 $mm\ a^{-1}$, $ts$ ~150 $a$ and $\dot{\varepsilon}$ ~ 1×10$^{-10}$ $mm\ a^{-1}$, $ts$ = 11860 $a$. The inversion results show that the IRSL50 profile of sample MBTP6 reaches steady state with erosion for an erosion rate $\dot{\varepsilon}$ = 4.3 ± 0.56 $mm\ a^{-1}$ (1$\sigma$) applied since at least 4 $a$. In these conditions, the corrected TCN age is $tc_{ss}$ = 6857 ± 991 $a$ (2.8% of correction). This steady state cannot be maintained for longer than 344 $a$ because further values correspond to the forbidden zone (Fig. 9d). The maximum corrected TCN age $tc_{max}$ = 68692 ± 10714 $a$ would represent a significant correction of 930%.

At steady state, the surfaces MBTP1 and MBTP6 would have lost 8.05 $mm$ and 17.2 $mm$ respectively. These values seem realistic regarding the natural surface textures observed on site: no smooth surface or striations are preserved on the roches moutonnées (Fig. 8). By taking the end-member hypothetical erosion values, the surfaces MBTP1 and MBTP6 would have lost maximum 102 $mm$ and 1479 $mm$ respectively.

**Table 3.** TCN and OSL surface ages and inversion results for samples MBTP1 and MBTP6

| Sample ID | TCN apparent age $t_0$ [1] | TCN age corr. $tc_{Css}$ [2] | TCN age corr. $tc_{Cmax}$ [2] | OSL surface exposure apparent age [3] | $t_S$ at SS* | $\dot{\varepsilon}$ at SS* | total erosion at SS* |
|---|---|---|---|---|---|---|---|
| | [a] | [a] | [a] | [a] | [a] | [mm a$^{-1}$] | [mm] |
| MBTP1 | 16428 ± 707 | 16619 ± 717 | 17396 ± 746 | 642 ± 160 | 2300 | 3.5 ± 1.2 × 10$^{-3}$ | 8.05 |
| MBTP6 | 6667 ± 965 | 6857 ± 991 | 68692 ± 10714 | 0.39 ± 0.02 | 4 | 4.3 ± 0.56 | 17.2 |

[a] Ages are calculated using the sea level high latitude (SLHL) rescaled local production rate of the Chironico landslide: 4.15 ± 0.10 $at\ g^{-1}\ a^{-1}$ rescaled for every longitude (Claude et al., 2014), latitude and elevation and considering no erosion correction, with the LSD scaling scheme (Lifton et al., 2014), the ERA40 atmospheric model (Uppala et al., 2005) and the Lifton VDM 2016 geomagnetic database (for ages in between 0-14 $ka$, Pavon-Carrasco et al., 2014 and for ages in between 14-75 $ka$, GLOPIS-75, Laj et al., 2004) by solving Eq. (3). (2) TCN age corr. $tc_{max}$ correspond to the maximum corrected TCN exposure ages calculating from the best maximum 5% solution. For (1) and (2) the errors represent the internal errors. (3) Ages were inverted (Fig. A5) using Eq. (2) and prescribing 10$^6$ solutions for a range of time from 0 to $t_0$ (TCN age calculated using the $^{10}$Be concentration of each sample and solving Eq. (3) without erosion correction). All models were calculated using the following parameters: $\overline{\sigma\varphi_0}$ = 129 $a^{-1}$, $\mu$ = 0.596 $mm^{-1}$, $D_0$ = 500 $Gy$, $\dot{D}$ = 7.4×10$^{-3}$ $Gy\ a^{-1}$ and $\dot{D}$ = 8.4×10$^{-3}$ $Gy\ a^{-1}$ respectively for sample MBTP1 and sample MBTP6. The uncertainties represent 1$\sigma$ of the distribution presented in Fig. A5. *SS means steady state.

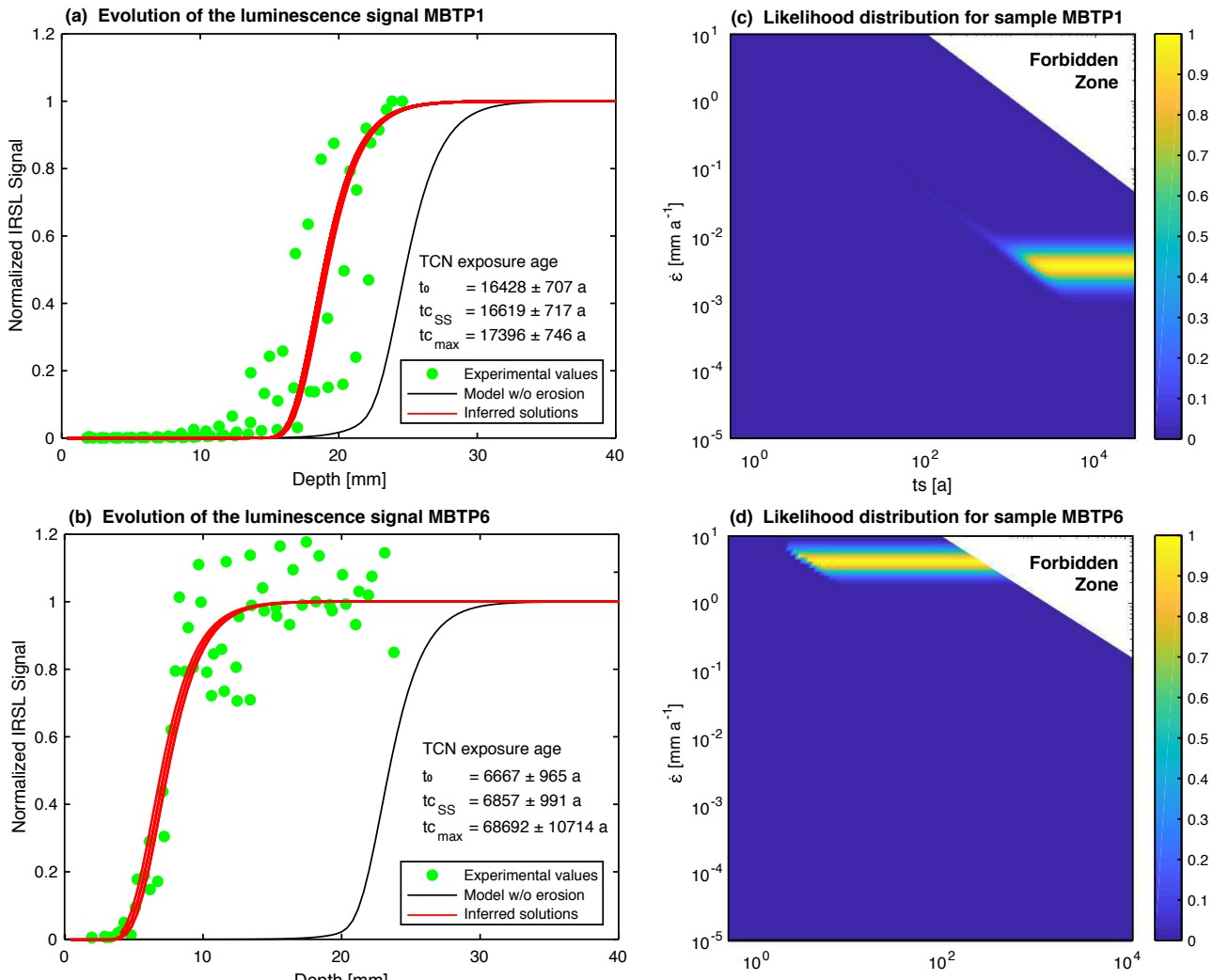

**. Figure 9.** IRSL50 profiles and inversion results for samples MBTP1 and MBTP6. (a) and (b) Green dots represent the measured IRSL50 profiles for samples MBTP1 and MBTP6 respectively. Black lines represent the reference profiles calculated using Eq. (2) and taking the TCN exposure age with no erosion correction ($t_0$). Red lines represent inferred fits where the likelihood is greater 0.95. $tc_{ss}$ represents the corrected TCN exposure age calculated at the steady state. $tc_{max}$ represents the maximum corrected TCN exposure age. (c) and (d) represent the likelihood distributions inverted from respective insets (a) and (b). All models were computed by solving Eq. (1) and using the following parameters: $\overline{\sigma\varphi_0} = 129\ a^{-1}$, $\mu = 0.596\ mm^{-1}$, $D_0 = 500\ Gy$, $\dot{D} = 7.4 \times 10^{-3}\ Gy\ a^{-1}$ and $\dot{D} = 8.4 \times 10^{-3}\ Gy\ a^{-1}$ for samples MBTP1 and MBTP6. Dose rates were calculated using the concentrations of U, Th, K and Rb of the bulk rock sample and the DRAC online calculator (details in Table A1; Durcan et al., 2015).

## 5 Discussion

The mismatch between OSL surface exposure and TCN ages presented in this study clearly show how significant the impact of erosion for OSL surface exposure dating is. If the luminescence bleaching front is interpreted without considering erosion, the resulting exposure age will be strongly underestimated (Figs. 5, 7 and 9). For samples MBTP1 and MBTP6 the apparent OSL surface exposure ages are $642 \pm 160$ $a$ and $0.32 \pm 0.02$ $a$, respectively while apparent TCN exposure ages are $16428 \pm 707$ $a$ and $6667 \pm 965$ $a$ respectively. We demonstrated in Section 2.1 that OSL surface exposure dating is hardly applicable to natural rock surfaces that experience even a minimal erosion rate of about $10^{-4}$ $mm$ $a^{-1}$. Our models and results show that the position of the bleaching front is highly sensitive to the erosion rate history. Recent studies (e.g., Freiesleben et al., 2015; Sohbati et al., 2012a, 2015; Rades el al., 2018) showed very convincingly that OSL rock surface dating can be used to identify multiple burial and exposure events in the history of a single clast. However, our results imply that erosion cannot be neglected. We show in this study that this high sensitivity to erosion can instead be used to estimate the erosion history of such rock surfaces.

To do so, we have numerically solved the equation describing the evolution of luminescence signal of a rock surface exposed to light and erosion (Eqs. (1) and (3)). The validation of the model was tested on synthetic data and applied to two different glacially-polished bedrock surfaces. We assumed a simple erosion rate history following a step function. However, it is very likely that rock surfaces are subject to stochastic erosion processes (e.g., Ganti et al., 2016). These stochastic processes cover potentially temperature, moisture, snow cover or wind fluctuations along the year. The numerical approach adopted here would potentially enable us to consider any type of erosion history (inverse exponential, stochastic distribution...). We considered the erosion rate to be non-constant in time but instead to follow a step function which changes from zero to a constant erosion rate at certain times of the exposure history. We observed that the resulting erosion histories can follow two states: a transient state or a steady state. Indeed, an experimental luminescence signal can be either at steady or transient state with erosion. To identify at which state the signal is, a model using Eq. (1) should try to fit the experimental luminescence signal considering a range of constant erosion rates applied over the TCN exposure age $t_0$ of the specific surface. If one specific erosion rate enables the model to fit the experimental luminescence signal, the system is at steady state with this specific erosion rate. If there is no unique solution, the system is at transient state with erosion. Note that some erosion rates cannot be applied for long durations. Indeed, the quantity of material removed and the concentration of cosmogenic nuclides in the rock surface would not match with the measured nuclide concentrations. To avoid that, we have defined a forbidden zone which characterized the range of pairs $\dot{\varepsilon}$ and $ts$ for which Eq. (3) could not be solved.

When a luminescence profile is derived from multiple erosion rate $\dot{\varepsilon}$ and time $ts$ pairs, the system is experiencing a transient state with erosion. This situation is characterized by a trade-off between erosion rate and the time of erosion. During this state, the luminescence signal does not evolve with depth if an increase of the erosion time is compensated by a decrease of the erosion rate. On the other hand, when a luminescence signal is derived from an erosion rate applied across a range of times $ts$, the system can be considered at steady state regarding the luminescence profile. In this case, the erosion rate can be considered as constant in time over the entire exposure age given by TCN dating providing that this solution falls outside of the forbidden

zone. At steady state, the time during which the erosion rate is applied is always lower or equal to the maximum corrected TCN age (i.e., $ts \leq tc_{max}$).

The luminescence profile from a given rock surface is able to give information about the erosion history of this surface at both transient and steady state with erosion. The coupling with TCN dating allows the determination of a limit in time of

the steady state with erosion, which cannot tend to infinity as discussed above (i.e., the forbidden zone). According to the inverse modeling of sample MBTP1, the total erosion experienced by the rock surface is about 8.05 $mm$ when the system reached steady state with erosion ($\dot{\varepsilon} = 3.5 \times 10^{-3}$ $mm$ $a^{-1}$ during $ts = 2300$ $a$) and 17.2 $mm$ for sample MBTP6 ($\dot{\varepsilon} = 4.3$ $mm$ $a^{-1}$ during $ts = 4$ $a$). This quantity of material removal is plausible given field observations, where the micro-structures of striations (coated layer and glacial polish) are not preserved but where the macro-patterns of glacial erosion can still be

observed (moulded forms, whalebacks, grooves). By taking the endmembers authorized by our model, we explore the limit of our method. The maximum total erosion is about 102 $mm$ for MBTP1 ($3.5 \times 10^{-3}$ $mm$ $a^{-1}$ during 29214 $a$) and about 1479 $mm$ for MBTP6 (4.3 $mm$ $a^{-1}$ during 344 $a$). Such high difference of erosion between two locations of the same vertical profile could be explain by the local topographic and environmental conditions such as slope surface and snow cover and controlling the efficiency of frost-cracking.

The quantification of the erosion rate distribution brings the opportunity to quantitatively correct TCN ages. These corrections can be minor but significant: for example about 1.1% for MBTP1 by taking the steady state values, about 5.8% using the endmember values. For sample MBTP6, the correction is about 2.8% by taking the steady state values. Using the endmember values, the maximum corrected TCN age for the highest sample is $tc_{max}$(MBTP1) = 17396 ± 746 $a$ and the lowest sample is $tc_{max}$(MBTP6) = 68692 ± 10714 $a$ (representing a maximum correction of about 930%). The assumption that a surface

at 2094 m.a.s.l. high (surface MBTP6) was exposed almost 50 ka longer than a surface located 451 meters higher (surface MBTP1 at 2545 m.a.s.l.) on the same vertical profile and in context of glacial thinning is hardly acceptable. According to the known glaciological evolution of Western Alps during LGM, exposure ages of > 25 $ka$ are simply not possible. Surfaces at 2600 m.a.s.l. located in accumulation zone of former glaciated area were covered by ice at least until the LGM (e.g., Penck and Brückner, 1909; Bini et al., 2009; Coutterand, 2010; Seguinot et al., 2018) which implies that the age estimates must be treated

with caution. However, our results imply that the uncertainty on the exposure age could be large. A correction of exposure age of few thousand years would have significant implications when investigating how post-LGM climate variability regionally impacted past ice extent.

We have presented the results using one luminescence signal only (IRSL50). Jenkins et al., (2018) and Sohbati et al. (2015) showed that multiple luminescence signals can be exploited. Since the bleaching propagates at different rates within the rocks

(c.f., Ou et al., 2018), using multiple signals (e.g., pIR225 and OSL125) should enable us to better assess whether the position of the bleaching front is steady or not and thus to further constrain the erosion history (both erosion rate and duration).

Our results confirm the results of Sohbati et al. (2018), who derived an analytical solution assuming steady erosion and using a confluent hypergeometric function. Here we solve the transient solution of Eq. (1) using the finite difference method. An important difference to the earlier study of Sohbati et al. (2018) is that here the system is fully coupled between OSL and

TCN surface exposure dating. OSL dating brings information about the evolution of the erosion rate in time and TCN dating give a realistic timeframe to this evolution by setting a forbidden zone.

The most striking outcome of this new approach is the ability to quantify surface erosion rates over timescales from 10 to $10^4$ $a$. The quantification of erosion rates using TCN concentration is limited (expressed in Sect. 2) with the minimum time given by $t >> 1/(\lambda + \nu \dot{\varepsilon})$. By taking the two endmembers of erosion of this study, $\dot{\varepsilon} = 10^{-5}$ $mm$ $a^{-1}$ and $\dot{\varepsilon} = 10$ $mm$ $a^{-1}$, the time limits are respectively $2 \times 10^6$ and $6 \times 10^4$ $a$ which means that one cannot use TCN to constrain the erosion history of post-LGM surfaces. Consequently, the coupling of OSL and TCN surface exposure dating makes the quantification of bare bedrock surface erosion possible at the timescale of a single interglacial event and might bring insight into the processes of topographic evolution in alpine environments.

## 6   Conclusions

In this study, we couple OSL and TCN surface exposure dating to constrain post-glacial bedrock erosion and surface exposure duration. We numerically solve the equation describing the evolution of luminescence signals in rock surfaces considering exposure age, bedrock surface erosion and the trapping and detrapping rates due to bleaching and athermal losses. We show that it is critical to account for bedrock surface erosion while interpreting luminescence bleaching profiles. Even at low erosion rates ($10^{-4}$ $mm$ $a^{-1}$) for periglacial environments, only few years are needed to affect the luminescence profile of a rock surface.

We were able to discriminate between two regimes characterizing the relationships between the depth of the luminescence bleaching, the exposure age and the bedrock surface erosion. The transient state describes a rock surface with a luminescence profile in disequilibrium. In contrast a rock surface in steady state is produced when the influence of bedrock surface erosion, exposure age and trapping rate compensate one another. If the system is maintained under these conditions, the luminescence signal no longer evolves with time. Indeed, the determination of the time at which the steady state with erosion occurs is critical. For the two natural surfaces we analyzed here, this time can range from 4 years (at an erosion rate of 4.3 $mm$ $a^{-1}$) to 2300 years (at an erosion rate of $3.5 \times 10^{-3}$ $mm$ $a^{-1}$). The approach developed in this study thus brings a new asset to directly quantify an erosion correction for TCN dating. We see that this correction can range from 1.2% to 930% for natural surfaces, although one must keep in mind that the exposure age may be overestimated if not compared to independent observations.

Finally, this new approach enables the quantification of erosion rates over surfaces exposed for $10$-$10^4$ $a$, filling a methodological gap in between short timescales (from few seconds to decades) and long-time scales ($> 10^5$ $a$). The contribution of this approach will allow quantification of the contribution of bare bedrock surface in sediment production and topographic evolution of alpine environments over glacial-interglacial cycles. Measurements in locations where bedrock surface erosion is very low (e.g., polar areas, high mountain) need to be investigated to check if OSL surface exposure is potentially applicable to timescale $> 10^2$ years without accounting for the effect of erosion rates. Another perspective is to investigate the control of temperature and climate on erosion rate evolution in time, along an elevation transect. Using this approach, the contribution of post-glacial bedrock erosion can be quantified and the feedback between erosion and climate evaluated.

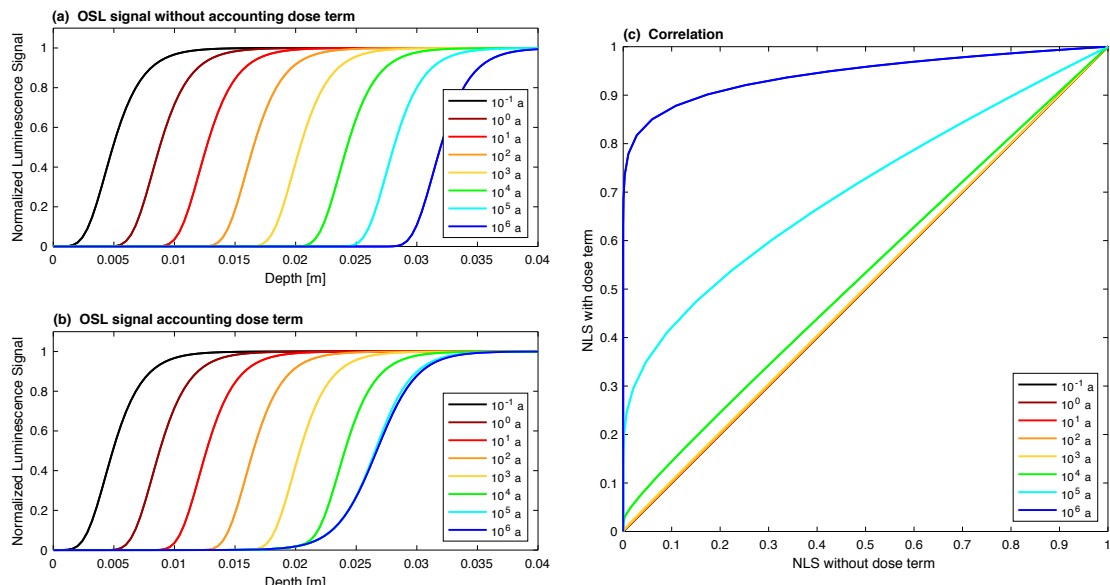

**Figure A1.** Modeled luminescence-depth profiles as predicted by Eq. (1) neither accounting for fading nor erosion and (a) without the trapping term and (b) with the trapping term, respectively. The selected parameter values are $\dot{D}= 8 \times 10^{-3} \ Gy \ a^{-1}$, $D_0= 500 \ Gy$, $\overline{\sigma\varphi_0}= 129$ $a^{-1}$ and $\mu = 0.596 \ mm^{-1}$. (c) is the comparison between the normalized luminescence (NLS) signal for both scenarios shown in (a) and (b).

**Table A1.** Dosimetry calculations for the feldspar samples analyzed. Conversion factors has been chosen after Adamiec and Aitken (1998). Alpha-particle attenuation and Beta-particle attenuation factors have been chosen after Bell (1980) and Mejdahl (1979) respectively. Cosmic dose rates have been calculated using the method of Prescott and Hutton (1994), assuming an overburden density of 2.7±0.1 g $cm^{-3}$. Internal K concentration is assumed to be 12±0.5% for both samples. Environmental dose rates were calculated using DRAC online calculator (Durcan et al., 2015), assuming a grain size between 750 and 1000 $\mu m$ and water content of 2%.

| Sample ID | U [ppm] | Th [ppm] | K [ppm] | Thickness [m] |
|-----------|---------|----------|---------|---------------|
| MBTP1 | 5.69 ± 0.12 | 36.8 ± 0.6 | 2.56 ± 0.03 | 0.08 ± 0.02 |
| MBTP6 | 8.75 ± 0.19 | 26.0 ± 0.4 | 3.88 ± 0.05 | 0.07 ± 0.02 |

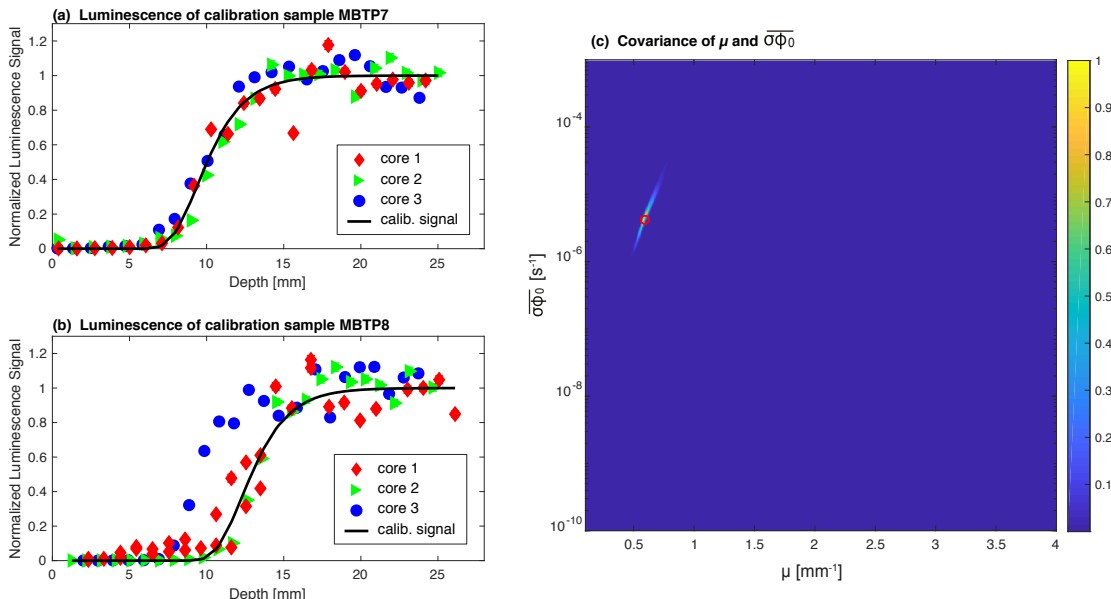

**Figure A2.** Calibration of the parameters $\mu$ and $\overline{\sigma\varphi_0}$ using two calibration samples MBTP7 (1936 m.a.s.l.) and MBPT8 (1995 m.a.s.l.) with exposure age of $2\pm2\ a$ and $11\pm2\ a$ respectively. These sampled were at the bottom of the Trélaporte vertical profiles in 2016. The surfaces are located between the present-day position of the glacier and the Little Ice Age maximal elevation. These ages were determined using the reconstruction from Vincent et al. (2014). The calibration is made through an inversion protocol by prediction $10^8$ luminescence signals corresponding to the combinations of $10^4$ values of $\overline{\sigma\varphi_0}$ in the logarithmic space and $10^4$ values of $\mu$. The inversed solutions are inferred using a least absolute deviation regression as described in Lehmann et al. (2018).

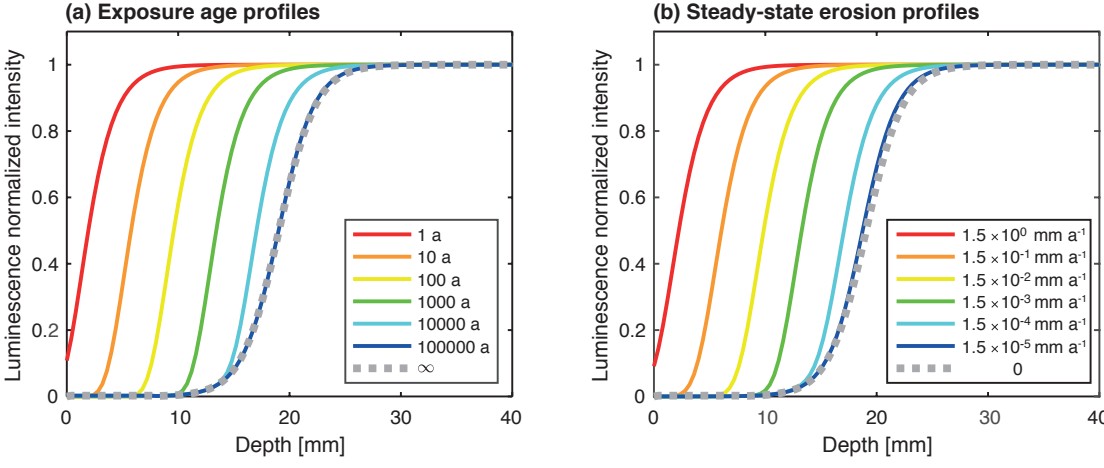

**Figure A3.** Modeled luminescence-depth profiles as predicted by Eq. (1) for a (a) non-eroding and (b) eroding rock surface, respectively. The selected parameter values are $\dot{D}=6\times10^{-3}\ Gy\ a^{-1}$, $D_0= 250\ Gy$, $\overline{\sigma\varphi_0} = 2200\ ka^{-1}$ and $\mu = 0.6\ mm^{-1}$ similar to Sohbati et al. (2018).

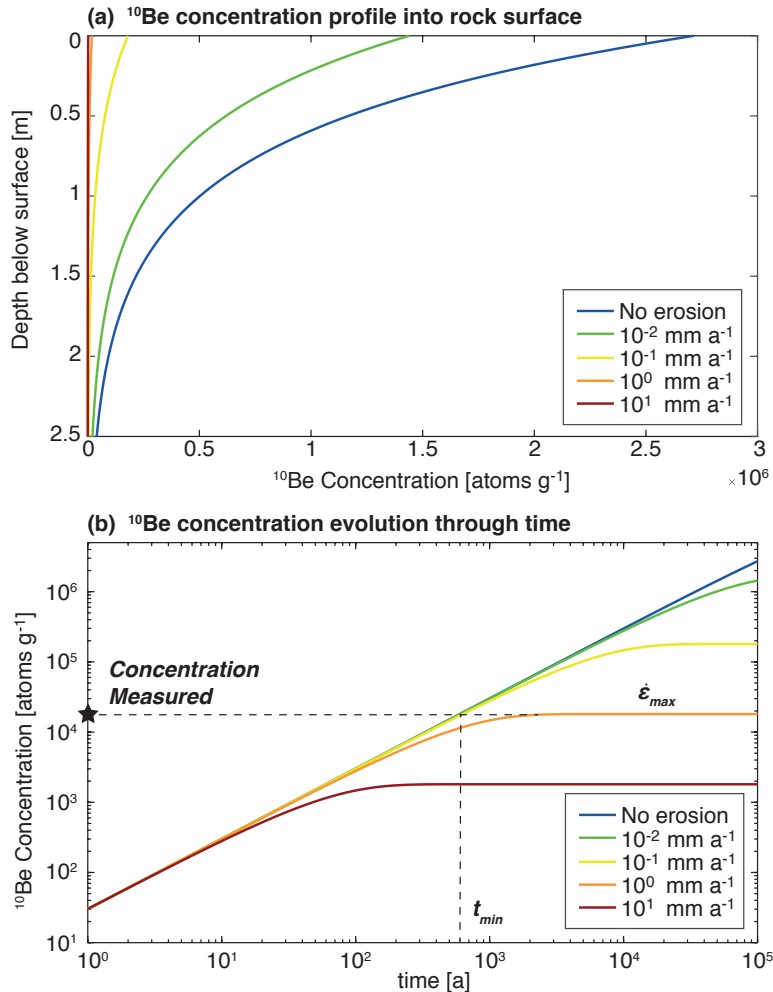

**Figure A4.** Evolution of the [10]Be production of a rock surface affected by different rates of erosion as a function of (a) the rock depth (b) the exposure age calculated using a modified version of the CREp online calculator to process non-linear erosion rate correction by solving Eq. (3) (Martin et al., 2017) as a modeling exercise and for comparison with OSL surface exposure curves of Fig. 5.

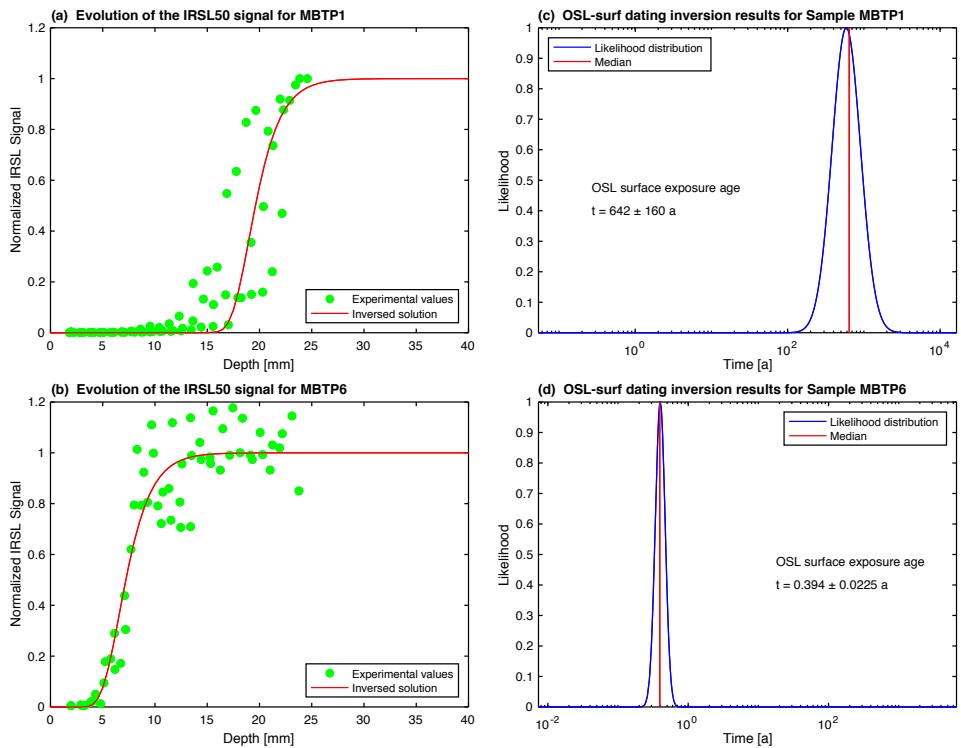

**Figure A5.** Determination of the apparent OSL surface exposure ages for samples MBTP1 and MBTP6. Experimental values in (a) and (b) correspond to the value measured for 3 cores per sample. The likelihood was determined using a probability density function following a least square deviations regression method minimizing the sum of the square differences between the experimental and the inverted values. Redlines in (c) and (d) represent the median value of the distribution. Apparent ages were inverted using Eq. (2) and prescribing $10^6$ solutions for a range of time from 0 to $t_0$ (TCN age calculated using the nuclide concentration of each sample and solving Eq. (3) without erosion correction). All models were calculated using the following parameters: $\overline{\sigma\varphi_0} = 129\ a^{-1}$, $\mu = 0.596\ mm^{-1}$, $D_0 = 500\ Gy$ and $\dot{D} = 7.4 \times 10^{-3}$ $Gy\ a^{-1}$ and $\dot{D} = 8.4 \times 10^{-3}\ Gy\ a^{-1}$ respectively for sample MBTP1 and sample MBTP6 (see main text for details).

**Table A2.** Infrared stimulated luminescence at 50°C (IRSL50) experimental values of sample MBTP1

| | | | | MBTP1 | | | | |
|---|---|---|---|---|---|---|---|---|
| | C1 | | | C2 | | | C3 | |
| x [mm] | Lx/Tx | Lx/Tx Err. | x [mm] | Lx/Tx | Lx/Tx Err. | x [mm] | Lx/Tx | Lx/Tx Err. |
| 1.81 | 0.00 | 0.000 | 2.24 | 0.00 | 0.000 | 1.97 | 0.00 | 0.005 |
| 2.80 | 0.00 | 0.000 | 3.16 | 0.00 | 0.001 | 2.91 | 0.00 | 0.001 |
| 3.76 | 0.00 | 0.001 | 4.14 | 0.00 | 0.001 | 3.96 | 0.00 | 0.000 |
| 4.70 | 0.00 | 0.001 | 5.09 | 0.00 | 0.001 | 4.99 | 0.00 | 0.001 |
| 5.72 | 0.00 | 0.001 | 6.07 | 0.00 | 0.001 | 5.95 | 0.00 | 0.001 |
| 6.80 | 0.00 | 0.001 | 7.10 | 0.00 | 0.000 | 6.85 | 0.00 | 0.002 |
| 7.77 | 0.00 | 0.002 | 8.04 | 0.00 | 0.001 | 7.72 | 0.01 | 0.003 |
| 8.68 | 0.00 | 0.002 | 8.89 | 0.00 | 0.001 | 8.62 | 0.01 | 0.007 |
| 9.52 | 0.00 | 0.001 | 9.77 | 0.00 | 0.002 | 9.54 | 0.03 | 0.013 |
| 10.49 | 0.01 | 0.003 | 10.72 | 0.01 | 0.002 | 10.42 | 0.02 | 0.004 |
| 11.53 | 0.01 | 0.002 | 11.70 | 0.01 | 0.003 | 11.36 | 0.04 | 0.022 |
| 12.49 | 0.01 | 0.002 | 12.64 | 0.02 | 0.008 | 12.32 | 0.07 | 0.011 |
| 13.47 | 0.01 | 0.006 | 13.63 | 0.05 | 0.010 | 13.65 | 0.19 | 0.109 |
| 14.41 | 0.02 | 0.018 | 14.63 | 0.13 | 0.175 | 15.00 | 0.24 | 0.073 |
| 15.56 | 0.02 | 0.014 | 15.60 | 0.11 | 0.032 | 15.95 | 0.26 | 0.100 |
| 17.02 | 0.03 | 0.005 | 16.76 | 0.15 | 0.072 | 16.88 | 0.55 | 0.193 |
| 18.25 | 0.14 | 0.176 | 17.93 | 0.14 | 0.127 | 17.79 | 0.63 | 0.109 |
| 19.24 | 0.15 | 0.149 | 19.19 | 0.36 | 0.091 | 18.73 | 0.83 | 0.171 |
| 20.30 | 0.16 | 0.108 | 20.38 | 0.50 | 0.101 | 19.65 | 0.87 | 0.150 |
| 21.23 | 0.24 | 0.179 | 21.29 | 0.74 | 0.125 | 20.82 | 0.79 | 0.165 |
| 22.16 | 0.47 | 0.348 | 22.30 | 0.88 | 0.118 | 21.98 | 0.92 | 0.136 |
| | | | 23.45 | 0.97 | 0.139 | 22.89 | 0.91 | 0.073 |
| | | | 24.59 | 1.00 | 0.082 | 23.86 | 1.00 | 0.082 |

**Table A3.** Infrared stimulated luminescence at 50°C (IRSL50) experimental values of sample MBTP6

| | MBTP6 | | | | | | | |
|---|---|---|---|---|---|---|---|---|
| | C1 | | | C2 | | | C3 | |
| x [mm] | Lx/Tx | Lx/Tx Err. | x [mm] | Lx/Tx | Lx/Tx Err. | x [mm] | Lx/Tx | Lx/Tx Err. |
| 1.96 | 0.00 | 0.000 | 1.96 | 0.01 | 0.000 | 3.32 | 0.01 | 0.000 |
| 3.00 | 0.01 | 0.000 | 2.90 | 0.01 | 0.000 | 4.30 | 0.05 | 0.001 |
| 4.05 | 0.02 | 0.001 | 3.84 | 0.02 | 0.000 | 5.25 | 0.18 | 0.004 |
| 5.11 | 0.09 | 0.002 | 4.80 | 0.01 | 0.000 | 6.17 | 0.15 | 0.003 |
| 6.13 | 0.29 | 0.007 | 5.76 | 0.19 | 0.004 | 7.09 | 0.44 | 0.010 |
| 7.19 | 0.30 | 0.008 | 6.72 | 0.17 | 0.004 | 8.00 | 0.79 | 0.017 |
| 8.29 | 1.01 | 0.022 | 7.71 | 0.62 | 0.013 | 8.93 | 0.92 | 0.020 |
| 9.29 | 0.81 | 0.017 | 8.69 | 0.79 | 0.017 | 9.85 | 1.00 | 0.021 |
| 10.27 | 0.79 | 0.019 | 9.68 | 1.11 | 0.024 | 10.76 | 0.85 | 0.018 |
| 11.34 | 0.86 | 0.019 | 10.61 | 0.72 | 0.016 | 11.67 | 1.12 | 0.024 |
| 12.39 | 0.81 | 0.020 | 11.53 | 0.73 | 0.018 | 12.58 | 0.96 | 0.021 |
| 13.40 | 1.14 | 0.025 | 12.46 | 0.71 | 0.016 | 13.50 | 0.99 | 0.021 |
| 14.29 | 1.04 | 0.023 | 13.40 | 0.71 | 0.017 | 14.41 | 0.97 | 0.021 |
| 15.26 | 0.98 | 0.023 | 14.49 | 1.23 | 0.026 | 15.33 | 0.96 | 0.021 |
| 17.48 | 1.28 | 0.028 | 15.56 | 1.16 | 0.025 | 16.25 | 0.93 | 0.021 |
| | | | 16.49 | 1.09 | 0.024 | 17.16 | 0.99 | 0.021 |
| | | | 17.45 | 1.18 | 0.025 | 18.14 | 1.00 | 0.021 |
| | | | 18.38 | 1.14 | 0.024 | 19.14 | 0.99 | 0.022 |
| | | | 19.32 | 0.97 | 0.021 | 20.07 | 1.08 | 0.023 |
| | | | 20.31 | 0.99 | 0.021 | 21.02 | 0.93 | 0.020 |
| | | | 21.26 | 1.03 | 0.022 | 21.93 | 1.02 | 0.022 |
| | | | 22.19 | 1.08 | 0.023 | 22.85 | 1.65 | 0.037 |
| | | | 23.11 | 1.14 | 0.025 | 23.78 | 0.85 | 0.024 |

*Competing interests.* TEXT

The authors declare having no competing interests.

*Acknowledgements.* This work was supported by the Swiss National Science Foundation (SNFS) funded Swiss-AlpArray SINERGIA project $(CRSII2-154434/1)$ and project $(PP00P2-170559)$ (P.G.V.). GEK acknowledges support from project $(Pz00P2-167960)$. The authors thank S. Ivy-Ochs, M. Christl, O. Kronig, E. Opyrchal, S. Casale and the Laboratory of Ion Beam Physics (LIP) at ETH Zürich for making the TCN dating preparation and analysis possible. The authors thank P.-H. Blard for sharing the code of the CREp calculator; D. Six and C. Vincent for GLACIOCLIM Alps data availability. We thank J. Braun for constructive input on the modeling. We thank S. Coutterand for his expertise of the Quaternary of the Mont-Blanc massif and his help during the sampling campaign. The authors would like to thank N. Stalder, J. González Holguera, G. Bustarret and U. Nanni for their support during field excursions M. Faria and K. Haring are thanked for laboratory support.

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
