# Peer review of "Evaluating post-glacial bedrock erosion and surface exposure duration by coupling in-situ OSL and 10Be dating"

_Earth Surface Dynamics, 2018_

## Referee Comment (RC1) · Anonymous Referee #1 · 3 Mar 2019

Lehmann et al present a novel way of constraining bedrock erosion rates by combining luminescence rock surface exposure dating (using the IR50 signal of feldspar) with cosmogenic radionuclide dating (10Be from quartz dissolution). They go through an intensive modelling effort and exploit the different but complementary spatial sensitivities that differ by an order of magnitude. In a given rock surface the buildup of 10Be is occurring in the top ca. 1-2 m, while the bleaching of the IR50 signal affects the topmost millimeters to centimeters only, making the luminescence rock surface exposure dating approach particularly sensitive to surface erosion.

The strength of this paper lies in the fact that Lehmann et al. recognize and sys-

tematically exploit these methodological differences. It thus represents an important contribution to the growing number of OSL rock surface dating studies and clearly shows (i) the limitation of the luminescence rock surface approach as a tool for purely obtaining exposure histories, particularly for older rock surfaces or environments with intensive surface erosion, (ii) opens up a way to check for the importance of erosion on a given rock surface and (iii) allows obtaining information on surface erosion. Lehmann et al show that their erosion rates from post LGM glacially polished rock surfaces obtained via their modelling and experimental data are sensible. Indeed, over millennial timescales such data are hardly obtainable via other techniques. This approach might also provide independent constraints for correcting terrestrial cosmogenic radionuclide ages. A note of caution: only two samples are included in the current study, and a more extensive dataset (both CRN and luminescence data) will be required to test the robustness of the modelling framework of Lehmann et al.

The main shortfall of the current version of the manuscript is the way the complex and interwoven modelling steps are presented. While many sections of the manuscript are clear and concise some other parts are hard to follow and in my opinion too brief, hence unclear and also sometimes inconsistent, particularly section 3.1. and the immediately following section 3.4 (sections 3.2 and 3.3 are missing or sections are mislabeled). Figures 6 and 7 could also be improved and linked with the text more intimately, thus improving the clarity of the presentation and comprehensibility of the modelling framework. I detail my main concerns in the following and append a list of smaller hiccups at the end.

Main issues – description and comprehensibility modelling steps and modelling framework (section 3):

p. 12, section 3: it would be helpful to define / explain the essence of the terms "forward model" and "inverse model" (e.g forward in time?) and the workflow in general terms before diving into details. This will help removing abstractness from your explanations. p. 13, section 3.1: please be more specific: first sentence "... a series of synthetic
luminescence profiles were generated ..." – refer to Figure and profiles (green dots, red lines, dotted lines, black lines?) What exactly is "a single experiment" – the generation of one synthetic luminescence profile? A set of modelling steps that result in Fig. 7a-d, respectively? How do your "experiments" differ from a "model" in line 21? Would it be better to talk about scenarios? These terms as well as the subsequent modelling steps and model setup are not always well defined yet. You go on in line 16: "In the first experiment ... (→ results in dashed line in Fig. 7a-d)" ... and in line 17: "In the second experiment ... " but what does this now result in? the green dots, the red curves in Fig. 7a-d? At the end of this paragraph you introduce the reference luminescence profiles (black lines) → would be helpful to move this upward and mention it together with e.g. constant erosion scenarios (dashed lines) before going into the more complex scenarios where erosion varies through time. Line 18: tc – is this the corrected TCN age? From Figs. 7a-d (text within figure) it looks like; but from Table 1 not necessarily so!? You introduce Fig. 7 in section 3.1 first; then you hop to Fig. 6 (that is unmentioned in the text up to this point) – this out of sequence move is a bit confusing. You have to elaborate on the concept of varying the erosion rate through time and on Fig. 6. The time axis in this figure needs to be read from right to left (because it is a forward model!?). The rational for using such step functions is not clear (here and in the related explanations on p. 9. L. 15) – what do you actually intend; to simulate climatic transitions e.g. from Pleistocene – Holocene in addition to capturing transient states? Sentence starting in line 18 onward: "Initially between... This is illustrated in Fig. 6" is unclear. Maybe you can improve Figure 6 (make a Fig. 6a and b out of it) and come up with a worked example illustrating how the scenarios in current Fig. 6 translate into a plot like Fig. 7a, b, c or d (which could become Fig. 6b?). In this context: my thinking was that the indicated values in fig. 7a-d (text within figure) for ts of 1 year and 100 years, respectively, should also be reflected in the tc versus the t0 ages (text within figure). Maybe this could be clarified with a Fig. 6a+b solution too. p. 14 line 6: it reads like the reference signal (what is the reference signal here? Black line in Fig. 7a-d? pls specify) is at 17 mm depth. But you actually mean

that the luminescence depth-profile is brought 7.8 mm closer to the surface relative to its former position, thus lying at 17mm absolute depth! Pls improve wording. Line 8: depth (instead of deep) line 9 onward to rest of this section: pls refer to figures whenever you actually discuss data/scenarios that are visualized in the respective Figures and thus link text and Figures much more closely than is currently the case! line 9: "... is applied for a duration of 1a" – unclear: does this mean that (referring to Fig. 6) the erosion only started 1 year (or 100 years in the case of fig. 7b and d) before sampling? i.e. for 16454 yrs no erosion; 1 year erosion of 1mm? – can this be integrated into a worked example (e.g. Fig. 6b, see above?) line 9: "...and integrated over its specific corrected exposure age" what exactly do you mean with integrated; with corrected for erosion 10-2 = tc max in Fig. 7a? pls specifiy p. 15 line 7: ts times 5x10-1 a = 182 days but in line 16 the same ts is 110 years!? Line 28: "which should be recovered in the inversion" = green dots in Fig. 7a-d? pls specify p. 16, line 7: ".... the would reproduce this specific lum signal (Fig. 7e)" = yellow triangle in Fig? pls specify Ad Fig. 7 e-h: what is the axis label of the colour bar at the right? What are the units?; green dots hard to see (better white?) Ad Fig. 7a-d: red lines – inferred solutions: are these the fits to the synthetic data? Pls explain in text; Fig. 7d: where is the dashed line – overlapping with red line? pls offset slightly. p. 18, line 5: by applying a constant erosion rate Line 8: what do you mean by insets here?; d and c have to be swapped.

Minor issues: p. 1 line 14: TCN abbreviation not explained here or in text p. 1, line 33: (Figs 1a and b) to a coarse-grained rough surface (Figs. 1c and 1d). p. 2 line 2-3: (e.g. deterioration .... Breakdown) – is this degree of detail really needed? You do not specify these terms and it thus remains unclear what the differences between these specific processes are... can be simplified. Line 10: erosion. Here you actually mean erosion of rock surfaces! Pls specify p. 3 line 15: burying them under sediment. No I think it was the other way round i.e. the sediment that is buried (sealed) due to large boulders (rock fall event) pls check. p. 3 last sentence: pls specify (cite) already here which papers / equations you are actually gonna review, because you start from established models. In the next and subsequent sentences you talk about

the proposed model – here you mean your own; or Sohbati or someone else? . . . so it is unclear what you are gonna review and how this will link with your own stuff. p.5 line 12: IRSL – it might be beneficial to briefly explain in the intro already that there are several signal that can be targeted depending on the mineral, rather than just hopping onto IR50 with preparing the reader for it. Line 12: Sentence: "Shobati et al., (2011, 2012a,b) introduced. . .duration". Is this the model you show below (Equation 1)? If yes pls specify (which paper?). In entire paragraph it is not quite clear from where equation 1 is taken from, of if you added some aspect to it!? And what Huntley's contribution to this specific equation exactly is. p. 5-6: suggestion: describe equation trems 1-4 first and explain Ou et al. + Sohbati's solution of equation therafter. p. 6 line 7: these parameters. Mü or what? p. 7 line 7: what is the fading term her in terms of g-value p. 9 line 19: NLS. Abbreviation not introduced in main text (only in fig caption) p. 20. Line 18: no figure 8 with IRSL curves in text of supplement! P 23. Line 22: show; ". . . that OSL-exposure can be sued to identify multiple burial and erosion events. . ." – but actually these approaches are not a pure OSL rock surface exposure approach but rather an OSL rock surface burial approach – which is not quite the same.

---

## Referee Comment (RC2) · Anonymous Referee #2 · 14 Mar 2019

This paper presents the first coupling of TCN and OSL surface exposure dating to quantify post-glacial erosion in paraglacial environments. The authors present sensitivity tests of a bleaching model and combine this model with a cosmogenic nuclide accumulation model to determine the erosion rates and durations that fit the measured data. The modelling is explained using synthetic data and is subsequently applied to two natural samples collected from a vertical profile along the Trélaporte ridge of the Mer de Glace glacier. The OSL technique deployed in this paper is very sensitive to erosion over short timescales. In the samples used here the thickness of rock removed ranges from 8.05 mm for sample MBTP1 ( $\delta IIJ \tilde{A} I \tilde{G} = 3.5 \times 10^{-3}$  mm a-1 for 2300 years) and 17.2 mm for sample MBTP6 ( $\delta IIJ \tilde{A} I \tilde{G} = 4.3$  mm a-1 for 4 years). The three

orders of magnitude variation in erosion rates cannot be reconciled with the geomorphology of the sample sites, and is not explored further in this paper. Overall the paper presents an exciting new approach for determining bedrock surface exposure ages and erosion rates using OSL. The theoretical coupling of OSL and TCN data is elegant but application to geological samples demonstrates that the results require very careful interpretation. The OSL technique deployed in this paper is very sensitive to erosion and scaling the results to longer term evolution of valley sides or even mountain ranges is likely to be to be difficult, as is clearly demonstrated by the geological samples used in this study. Nonetheless, the approach is very promising.

Specific suggested changes:

p1, line 29: 'Glacially-polished bedrock, or so-called "roche moutonées" offers', not all glacially polished rock is a roche moutonée, for example Fig 1c, please change to 'Glacially polished bedrock offers' p2, line 15: 'of know age' should be 'of known age' p3, line 1: 'until being completely' should be 'until completely' p3, line 8: add 'a' after ~106 p3, line 13: 'historical' could be 'historic' p3, line 14: 'in the Canyonlands' should be 'in Canyonlands' p4, line 4: change 'roche moutonée' to glacially polished bedrock p4, line 6: delete 'a' before transient p4, line 11-14, Fig 1 caption: where is the craig and tail referred to in the caption? Roches moutonée are not short-lived features, neither are crag and tails. Fig 1c does not show roches moutonée morphology as stated in the caption. It shows glacially abraded. Roches moutonée have guite specific morphology. p6, line 20-29: these two paragraphs explaining the 3rd and 4th terms of eq. 1 should be placed before the para starting with 'Ou et al. (2018)...' on p5, line 34. p6, line 29: you state that you 'obtain exactly the same results using our numerical solution (Fig. A3).' Where is this demonstrated. Fig. A3 does not show a comparison between Sohbati et al. (2018) and your work. It would be good to show how 'exactly the same' your results are. p.8, line 18: please explain the '15 and 25 mm values for our end-member simulations (Fig. 4).' The values do not appear to match the curves in the figure. p.9, Fig. 4 caption: 'Sect. 2.1.2' should be 'Sect.

**ESurfD**
2.1.1' p.10, Fig. 5 caption: 'Sect. 2.1.2' should be 'Sect. 2.1.1'. Please check all occurrences of cross-referencing carefully. p.12, line 6: 'samples used in the following of this study (Table 3).' Delete 'the following'. Also, Table 3 does not show the averages for D-dot or D-zero. Which table are you referring to? p.13, line 2-3: the erosion rates 10-2 mm a-1 and 1mm a-1 do not appear in Sect. 2.1.2 as stated. p.14, line 8: 'Figs. 6a, b, c, d' should be 'Figs. 7a, b, c, d' p.14, line 17: delete 'but constant for an infinite' p.15, line 3: 'valid' should be 'validate' p.15, line 7: check Sect number p.15, line 7-8: 'this range being arbitrarily decided even so the upper boundary is set to be approximately' should be 'this range being arbitrarily decided with the upper boundary set to approximately p.15, line 12: 'parameters' should be 'parameter' p.15, line 15: delete 'further the limit laying in' p.18, Fig 8c: show the precise location of sample MBTP6. This is important to explain the shielding value in Table 3. p.20, Table 2 caption, line 8: 'in between' should be 'between'. This happens twice in the line p.20, line 13-16: how is it possible that the calculated to exposure age uncertainties are smaller than the measured cosmogenic nuclide concentration uncertainties. p.20, line 18: 'Figure 8' should be 'Figure 9' p.20, line 30: 'reference profile is lying at 23.5 mm' should be 'reference profile is at 23.5 mm' p.21, line 9: 'lies in between' should be 'lies between' p.21, line 9: 'e = 1 mm  $a^{-1}$ ' should be 'e = 10 mm  $a^{-1}$ ' p.21, line 10: 'Sect. 3.2' should be 'Sect. 3.3' p.21, line 15: ' $e = 1 \text{ mm a}^{-1}$ ' should be ' $e = 1 \text{ mm}^{-1}$ ' should be 'e 10 mm a-1' p.23, line 20: 'erosion rate about' should be 'erosion rate of about' p.23, line 22: '(Rades el al. 2018) have showed' should be '(Rades et al. 2018) showed' p.24, line 9: 'for too long duration' should be 'for long durations' p.24, line 12: 'time ts pair' should be 'time ts pairs' p.24, line 24: '( $\delta$ IJAÌG = 4.3 m a^-1 during ts = 4 years)' should be '(ðIJJAIG = 4.3 mm a^-1 during ts = 4 years)', i.e. millimetres, not metres p.24, line 27: 'limit our method' should be 'limits of our nethod' p.24, line 29: 'Such high difference of erosion between two locations of the same vertical profile is unlikely'. I think this statement is not supported by your data. Considering the difference in sample shielding it appears that MBTP6 was collected from a steeper slope than MBTP1. Fig. 3 suggests that the rock face may have lost mass by spallation, which could explain

**ESurfD**
the order of magnitude lower 10Be concentration. These types of issues should be explored more. p.24, line 34: 'The assumption that surface at 2094....almost 50ka latter than...' should be 'The assumption that a surface at 2094....almost 50ka longer than...' p.25, line 1: 'latter' should be 'later' p.26, line 5: 'the correction TCN dating of erosion' should be 'an erosion correction for TCN dating' p.26, line 9: 'gab' should be 'gap' p.28, Fig. A2 caption: 'These samples were in 2016 ....profiles' should be 'These samples were...profiles in 2016' p.28, Fig. A3 caption: 'comparable to the average values obtained...' What does comparable mean? What were the average values? Quantify "comparable". p.29, Fig. A4 caption: 'exposure age obtains using' should be 'exposure age calculated using'

**ESurfD**

---

## Author Comment (AC1) · 6 May 2019

Answers to Anonymous Referee #1

Lehmann et al. present a novel way of constraining bedrock erosion rates by combining luminescence rock surface exposure dating (using the IR50 signal of feldspar) with cosmogenic radionuclide dating ($^{10}$Be from quartz dissolution). They go through an intensive modelling effort and exploit the different but complementary spatial sensitivities that differ by an order of magnitude. In a given rock surface the buildup of $^{10}$Be is occurring in the top ca. 1-2 m, while the bleaching of the IR50 signal affects the topmost millimeters to centimeters only, making the luminescence rock surface exposure dating approach particularly sensitive to surface erosion.

The strength of this paper lies in the fact that Lehmann et al. recognize and systematically exploit these methodological differences. It thus represents an important contribution to the growing number of OSL rock surface dating studies and clearly shows (i) the limitation of the luminescence rock surface approach as a tool for purely obtaining exposure histories, particularly for older rock surfaces or environments with intensive surface erosion, (ii) opens up a way to check for the importance of erosion on a given rock surface and (iii) allows obtaining information on surface erosion. Lehmann et al. show that their erosion rates from post LGM glacially polished rock surfaces obtained via their modelling and experimental data are sensible. Indeed, over millennial timescales such data are hardly obtainable via other techniques. This approach might also provide independent constraints for correcting terrestrial cosmogenic radionuclide ages. A note of caution: only two samples are included in the current study, and a more extensive dataset (both CRN and luminescence data) will be required to test the robustness of the modelling framework of Lehmann et al.

The main shortfall of the current version of the manuscript is the way the complex and interwoven modelling steps are presented. While many sections of the manuscript are clear and concise some other parts are hard to follow and, in my opinion, too brief, hence unclear and also sometimes inconsistent, particularly section 3.1. and the immediately following section 3.4 (sections 3.2 and 3.3 are missing or sections are mislabeled). Figures 6 and 7 could also be improved and linked with the text more intimately, thus improving the clarity of the presentation and comprehensibility of the modelling framework. I detail my main concerns in the following and append a list of smaller hiccups at the end.

Main issues – description and comprehensibility modelling steps and modelling framework (section 3):

**We are grateful for this very constructive review provided by Anonymous Referee #1. The presentation and the comprehensibility of the modelling steps (section 3) have significantly improved. We thank the reviewer to point out the numbering problem, which has been fixed in the new version of the manuscript. Please find in the following the answer and comment on the reviewer feedbacks. Comments of the reviewer are underling and our answers are in bolt.**

p. 12, section 3: it would be helpful to define / explain the essence of the terms "forward model" and "inverse model" (e.g forward in time?) and the workflow in general terms before diving into details. This will help removing abstractness from your explanations.
**We added the following note at the beginning of the Section 3, page 12: "In this section, we generate a series of forward and inverse models. The forward model calculates a luminescence signal and a $^{10}$Be concentration from synthetic erosion and exposure histories. The goal of the inverse model is to constrain the model parameters (i.e., erosion and exposure histories) using the data (i.e., IRSL signal and $^{10}$Be concentration). To validate the inversion procedure, we use the forward model to create synthetic data which we then recover using the inverse model."**

p. 13, section 3.1: please be more specific: first sentence "… a series of synthetic luminescence profiles were generated…" – refer to Figure and profiles (green dots, red lines, dotted lines, black lines?)

The sentence: "The first experiment assumes a constant erosion rate over the TCN exposure age $t_S = t_0$" was changed to "For this scenario, erosion rates are assumed to be constant over the TCN exposure age $t_S = t_0$.". We also specifically refer to "dashed lines in Figs. 7a-d".

What exactly is "a single experiment" – the generation of one synthetic luminescence profile? A set of modelling steps that result in Fig. 7a-d, respectively?
**In the third scenario, another set of synthetic luminescence profiles were again generated using Eq. (1) in a forward model, but the erosion rate was allowed to vary with time (green dots in Figs. 7a-d). We have rephrased this statement accordingly.**

How do your "experiments" differ from a "model" in line 21?
**Those are the same experiments. The sentence was changed to "We report the four model outputs calculated using $t_S$ between 1 and 100 a, and erosion rates $\dot{\varepsilon}$ between $10^{-2}$ and 1 mm $a^{-1}$ (green dots respectively in Figs. 7a-d)."**

Would it be better to talk about scenarios?
**For clarity "experiment" was change for "scenario".**

These terms as well as the subsequent modelling steps and model setup are not always well defined yet. You go on in line 16: "In the first experiment… ($\rightarrow$ results in dashed line in Fig. 7a-d)" … and in line 17: "In the second experiment …" but what does this now result in? the green dots, the red curves in Fig. 7a-d?
**The mention "(green dots in Figs. 7a-d)" was added at the end of this sentence.**

At the end of this paragraph you introduce the reference luminescence profiles (black lines) $\rightarrow$ would be helpful to move this upward and mention it together with e.g. constant erosion scenarios (dashed lines) before going into the more complex scenarios where erosion varies through time.
**Changed as suggested.**

Line 18: tc – is this the corrected TCN age? From Figs. 7a-d (text within figure) it looks like; but from Table 1 not necessarily so!?
**tc is indeed the corrected TCN age. We moved $t_0$ and $t_C$ from the "Both methods" section to the "TCN dating" section of Table 1, and include TCN in front of "exposure age".**

You introduce Fig. 7 in section 3.1 first; then you hop to Fig. 6 (that is unmentioned in the text up to this point) – this out of sequence move is a bit confusing.
**We added a mention of Figure 6 in the paragraph 3.1. This sentence was added in this paragraph: "Figure 6 illustrates the schematic representation of four different erosion scenarios through time (Figs. 6a and 6b) and their resulting luminescence signal (Fig. 6c)."**

You have to elaborate on the concept of varying the erosion rate through time and on Fig. 6. The time axis in this figure needs to be read from right to left (because it is a forward model!?).
**Figure 6 was flipped to have the exposure event on the left and so the time going from left to right. The purpose of this figure was to illustrate the step function used an erosion rate history. We hope that with this updated figure the reader understands more the concept of step function in time.**

The rational for using such step functions is not clear (here and in the related explanations on p. 9. L. 15) – what do you actually intend; to simulate climatic transitions e.g. from Pleistocene – Holocene in addition to capturing transient states? Sentence starting in line 18 onward: "Initially between… This is illustrated in Fig. 6" is unclear.
**"The assumption made here, is that the evolution of erosion in time can follow a step function. Our objective is to explore the effect of a non-constant erosion rate in time on both the luminescence signal and $^{10}$Be concentration. This is the simplest possible time varying erosion rate history. The erosion is initially equal to zero, i.e., between the corrected exposure age tc, and an**

**onset time of erosion ts, and increase to a fixed rate between ts and today. Note that more sophisticated erosion rate histories could be tested with the same approach, which is beyond the scope of the current study."**

Maybe you can improve Figure 6 (make a Fig. 6a and b out of it) and come up with a worked example illustrating how the scenarios in current Fig. 6 translate into a plot like Fig. 7a, b, c or d (which could become Fig. 6b?). In this context: my thinking was that the indicated values in Fig. 7a-d (text within figure) for ts of 1 year and 100 years, respectively, should also be reflected in the tc versus the t0 ages (text within figure). Maybe this could be clarified with a Fig. 6a+b solution too.

**Figure 6 was changed and divided in Figure 6a being a flip version was flip to have the exposure event on the left and so the time going from left to right. The purpose of this figure was to conceptualize how the step function is set. We hope that with this new figure the reader understands more the concept of step function in time. Figure 6b illustrates how erosion history can be described by a single dot in a diagram "Erosion rate $\dot{\varepsilon}$" vs "Onset erosion time $t_S$". Finally, Figure 6c presents the resulting luminescence signal for the different erosion histories. In this way, Figure 6 gives a clear explanation of how experiments were designed and prepares the reader for Fig. 7. "Schematic representation of four different erosion scenarios through time (a) and (b) and their resulting luminescence signal (c). $t_0$ is the uncorrected $^{10}$Be exposure age, $t_S$ the onset times of erosion, $t_C$ the corrected exposure ages, and $\dot{\varepsilon}$ the erosion rate. Note that the luminescence plots in (c) are not model outputs but drawings, with the aim of conceptualizing how the experiments are designed."**

p. 14 line 6: it reads like the reference signal (what is the reference signal here? Black line in Fig. 7a-d? pls specify) is at 17 mm depth. But you actually mean that the luminescence depth-profile is brought 7.8 mm closer to the surface relative to its former position, thus lying at 17 mm absolute depth! Pls improve wording. Line 8: depth (instead of deep)

**This paragraph was rearranged in the following way:**
**"By applying a constant erosion rate of $10^{-2}$ mm a$^{-1}$ to a rock surface exposed since $t_0$ (16428 ± 589 a), the luminescence signal is brought 7.8 mm closer to the surface (i.e., 17 mm deep from the surface) compared to the reference signal (luminescence signal exposed since $t_0$ and no affected by erosion; black line in Figs. 7a-d at 24.8 mm deep from the surface). For a constant erosion rate of 1 mm a$^{-1}$, the luminescence signal is brought 15.4 mm closer to the surface (i.e., 9.4 mm deep from the surface) compared to the reference signal (difference between black lines and dash lines measured at NLS = 0.5 in Figs. 7a-d).**

Line 9 onward to rest of this section: pls refer to figures whenever you actually discuss data/scenarios that are visualized in the respective Figures and thus link text and Figures much more closely than is currently the case!

**We thank the reviewer to point this lack of clarity, the referring to figures was improved as suggested.**

Line 9: "…is applied for a duration of 1a" – unclear: does this mean that (referring to Fig. 6) the erosion only started 1 year (or 100 years in the case of fig. 7b and d) before sampling? i.e. for 16454 yrs no erosion; 1 year erosion of 1mm? – can this be integrated into a worked example (e.g. Fig. 6b, see above?)

**The durations of erosion (1 and 100 years) correspond to a duration before sampling. We add the mention "before sampling" after each duration in the text.**

Line 9: "…and integrated over its specific corrected exposure age" what exactly do you mean with integrated; with corrected for erosion $10^{-2}$ = tc max in Fig. 7a? pls specifiy

**This sentence was changed for: "and for an exposure time corrected with its specific erosion history tc".**

p. 15 line 7: ts times $5 \times 10^{-1}$ a = 182 days but in line 16 the same ts is 110 years!?

The ts values mentioned in line 7 are the values defining the space in which the inversion model will be sampled to reproduce the experimental / synthetic values (i.e., $5 \times 10^{-1}$ a and $3 \times 10^4$ a). However, the values mentioned in line 16 are the values defining the space where the $^{10}$Be concentration will not be possible to recover due to too strong erosion and described as the "forbidden zone". This space is defined with the following boundary conditions $\dot{\varepsilon} = 10$ mm a$^{-1}$, $t_S$ ~110 a and $\dot{\varepsilon} \sim 5 \times 10^{-1}$ mm a$^{-1}$, $t_S = 29210$ a.

Line 28: "which should be recovered in the inversion" = green dots in Fig. 7a-d? pls specify
**We thank the reviewer to point this lack of clarity, the referring was added as suggested.**

p. 16, line 7: "... the would reproduce this specific lum signal (Fig. 7e)" = yellow triangle in Fig? pls specify
**We thank the reviewer to point this lack of clarity, the mention: "Normalized likelihood > 0.9: yellow area in Fig. 7e" was added.**

Ad Fig. 7 e-h: what is the axis label of the colour bar at the right? What are the units?; green dots hard to see (better white?)
**The unit of the colour bar at the right was added "Normalized Likelihood", yellow (value 1) means high probability to have recovered the solution. Green and black dots were changed for green and black strokes and white fills.**

Ad Fig. 7a-d: red lines – inferred solutions: are these the fits to the synthetic data? Pls explain in text;
**The sentence on page 15, line 30 was improved in this way: "We then select the pairs of $\dot{\varepsilon}$ and $t_S$ leading to the maximum 5% likelihood values which are fitting the synthetic data (the threshold of 5% is arbitrarily chosen), and plot their corresponding luminescence profile values (red lines in Figs. 7a-d)."**

Fig. 7d: where is the dashed line – overlapping with red line? pls offset slightly.
**Changed as suggested.**

p. 18, line 5: by applying a constant erosion rate
**Changed as suggested.**

Line 8: what do you mean by insets here?; d and c have to be swapped.
**The sentence has been changed to "(e), (f), (g) and (h) represents the likelihood distributions inverted from the synthetic luminescence profiles respectively in (a), (b), (c) and (d)."**

**Minor issues**

p. 1 line 14: TCN abbreviation not explained here or in text
**Changed "in situ cosmogenic $^{10}$Be (TCN)" to "terrestrial cosmogenic nuclide $^{10}$Be (TCN)" in both the abstract and the main text.**

p. 1, line 33: (Figs 1a and b) to a coarse-grained rough surface (Figs. 1c and 1d).
**We thank the reviewer to point this mistake, we changed as suggested.**

p. 2 line 2-3: (e.g. deterioration … Breakdown) – is this degree of detail really needed? You do not specify these terms and it thus remains unclear what the differences between these specific processes are… can be simplified.
**"(e.g., deterioration, decay, crumbling, decomposition, rotting, disintegration, disaggregation or breakdown)" was deleted for simplicity.**

Line 10: erosion. Here you actually mean erosion of rock surfaces! Pls specify
**"of rock surfaces" was added after "erosion" as suggested.**

p. 3 line 15: burying them under sediment. No I think it was the other way round i.e. the sediment that is buried (sealed) due to large boulders (rock fall event) pls check.
**This part has been changed to the following: "Some of the paintings were damaged by a rockfall event, and conventional luminescence was applied on a rockfall boulder and buried sediments (Chapot et al., 2012). This provided a minimum age for the event."**

p. 3 last sentence: pls specify (cite) already here which papers / equations you are actually gonna review, because you start from established models. In the next and subsequent sentences you talk about the proposed model – here you mean your own; or Sohbati or someone else? …so it is unclear what you are gonna review and how this will link with your own stuff.
**We thank the reviewer to point this lack of clarity. As the reviewer mentioned we don't review papers / equations but we start from establishing models. In this sense, the sentence "To achieve this, we first review the theoretical and model approach to simulate the evolution of luminescence signals in rock surfaces" was removed. And the next sentence was modified in this manner: "To achieve this, we developed a new model which depends on the exposure age, the surface erosion, the trapping and detrapping (bleaching) rates and the athermal loss (c.f. Eq. 1, Section 2.1.1)".**

p.5 line 12: IRSL – it might be beneficial to briefly explain in the intro already that there are several signal that can be targeted depending on the mineral, rather than just hopping onto IR50 with preparing the reader for it.
**Following this suggestion, we added that statement at the end of the introduction (page 3, line 20): "Note that several signals can be targeted in the same rock slice depending on the mineral (e.g., Sohbati et al., 2015; Jenkins et al., 2018). OSL is usually used to analysed the luminescence of quartz (Murray and Wintle, 2000) and IRSL for potassium-rich feldspar signal (both at 50°C and 225°C, Buylart et al., 2009)."**

Line 12: Sentence: "Shobati et al., (2011, 2012a,b) introduced … duration". Is this the model you show below (Equation 1)? If yes pls specify (which paper?). In entire paragraph it is not quite clear from where equation 1 is taken from, of if you added some aspect to it!?
**Equation 1 is the new model proposed in this actual study: This section was changed to: "Sohbati et al. (2011, 2012a, b) introduced a mathematical model that describes the process of luminescence bleaching with depth in a homogeneous lithology, enabling the quantification of rock surface exposure duration. Here we propose a new model describing the evolution of luminescence in rock surface as a function of different parameters characterizing the probability of charge trapping […] (c.f. Eq.1, Section 2.1.1)."**

And what Huntley's contribution to this specific equation exactly is.
**Huntley's contribution was mentioned after the parameter r' for recombination center distance. We acknowledged this was not relevant, and thus removed the citation and mentioned it later when we go into details of the fading parameters.**

p. 5-6: suggestion: describe equation terms 1-4 first and explain Ou et al. + Sohbati's solution of equation therafter.
**Changed as suggested.**

p. 6 line 7: these parameters. µ or what?
**We thank the reviewer to point this lack of clarity, we changed the sentence for "[…] for a complete description of $\overline{\sigma\varphi}_0$ and µ parameters […]".**

p. 7 line 7: what is the fading term her in terms of g-value
**The values mentioned in the text correspond to two end-members between g-values ~0%/decade and 10-15%/decade.**

p. 9 line 19: NLS. Abbreviation not introduced in main text (only in fig caption)
**We checked and NLS is introduced on page7, line 12.**

p. 20. Line 18: no figure 8 with IRSL curves in text of supplement!
**We thank the reviewer to point this mistake, figures were wrongly labeled. This has been corrected.**

p 23. Line 22: show; "… that OSL-exposure can be used to identify multiple burial and erosion events…" – but actually these approaches are not a pure OSL rock surface exposure approach but rather an OSL rock surface burial approach – which is not quite the same.
**We thank the reviewer to point this mistake, the studies mentioned in the text are indeed using OSL rock surface dating, which include both exposure and burial. But this is just a word issue, the exposure problem (erosion) is still valid for that approach if one wants to recover exposure durations. The term "OSL-exposure" was changed for "OSL rock surface dating".**

---

## Author Comment (AC2) · 6 May 2019

Answers to Anonymous Referee #2

This paper presents the first coupling of TCN and OSL surface exposure dating to quantify post glacial erosion in paraglacial environments. The authors present sensitivity tests of a bleaching model and combine this model with a cosmogenic nuclide accumulation model to determine the erosion rates and durations that fit the measured data. The modelling is explained using synthetic data and is subsequently applied to two natural samples collected from a vertical profile along the Trélaporte ridge of the Mer de Glace glacier. The OSL technique deployed in this paper is very sensitive to erosion over short timescales. In the samples used here the thickness of rock removed ranges from 8.05 mm for sample MBTP1 ($\dot{\varepsilon} = 3.5 \times 10^{-3}$ mm a$^{-1}$ for 2300 years) and 17.2 mm for sample MBTP6 ($\dot{\varepsilon} = 4.3$ mm a$^{-1}$ for 4 years). The three orders of magnitude variation in erosion rates cannot be reconciled with the geomorphology of the sample sites, and is not explored further in this paper. Overall the paper presents an exciting new approach for determining bedrock surface exposure ages and erosion rates using OSL. The theoretical coupling of OSL and TCN data is elegant but application to geological samples demonstrates that the results require very careful interpretation.

The OSL technique deployed in this paper is very sensitive to erosion and scaling the results to longer term evolution of valley sides or even mountain ranges is likely to be to be difficult, as is clearly demonstrated by the geological samples used in this study. Nonetheless, the approach is very promising.

**We are grateful for this very constructive review provided by Anonymous Referee #2. Please find in the following the answer and comment on the reviewer feedbacks. Comments of the reviewer are underling and our answers are in bold.**

p1, line 29: 'Glacially-polished bedrock, or so-called "roche moutonées" offers', not all glacially polished rock is a roche moutonée, for example Fig 1c, please change to 'Glacially polished bedrock offers'
**Changed as suggested.**

p2, line 15: 'of know age' should be 'of known age'
**Changed as suggested.**

p3, line 1: 'until being completely' should be 'until completely'
**Changed as suggested.**

p3, line 8: add 'a' after $10^6$
**Changed as suggested.**

p3, line 13: 'historical' could be 'historic'
**Changed as suggested.**

p3, line 14: 'in the Canyonlands' should be 'in Canyonlands'
**Changed as suggested.**

p4, line 4: change 'roche moutonée' to glacially polished bedrock
**Changed as suggested.**

p4, line 6: delete 'a' before transient
**Changed as suggested.**

p4, line 11-14, Fig 1 caption: where is the craig and tail referred to in the caption? Roches moutonée are not short-lived features, neither are crag and tails.

**In the sake of clarity, we removed "roches moutonées, craig and tails" from the caption.**

Fig 1c does not show roches moutonée morphology as stated in the caption. It shows glacially abraded. Roches moutonée have quite specific morphology.
**We changed "roches moutonnée" to "glacially abraded surfaces".**

p6, line 20-29: these two paragraphs explaining the 3rd and 4th terms of eq. 1 should be placed before the para starting with 'Ou et al. (2018)...' on p5, line 34.
**Changed as suggested.**

p6, line 29: you state that you 'obtain exactly the same results using our numerical solution (Fig. A3).' Where is this demonstrated. Fig. A3 does not show a comparison between Sohbati et al. (2018) and your work. It would be good to show how 'exactly the same' your results are.
**Figure A3 shows the output of the model using the same parameter of Sohbati et al. (2018) study. The comparison between the two approaches is made by visual inspection of the shape of the bleaching front and the depth $x_{50\%}$ defined as NLS($x_{50\%}$)=0.5 value for every model outputs (NLS = Normalized luminescence signal). We changed the sentence with "we obtain results which are similar to their results calculated using their an analytical solution (Fig. A3)."**

p.8, line 18: please explain the '15 and 25 mm values for our end-member simulations (Fig. 4).' The values do not appear to match the curves in the figure.
**We thank the reviewer to point this lack of clarity, we changed for "22 and 31 mm (measured at the inflection point)".**

p.9, Fig. 4 caption: 'Sect. 2.1.2' should be 'Sect. 2.1.1'
**We kept Sect. 2.1.2. because in this section it is mentioned the "We use $\overline{\sigma\varphi_0}$ = 129 a$^{-1}$ and μ = 0.596 mm$^{-1}$ that were determined from two calibration rock surfaces of similar granitic lithology from the Mont Blanc massif, with no erosion and known exposure age (Fig. A2). The values $\dot{D}$ = 8 Gy ka$^{-1}$ and $D_0$ = 500 Gy were selected as they are comparable to the average values obtained for samples used in this study."**

p.10, Fig. 5 caption: 'Sect. 2.1.2' should be 'Sect. 2.1.1'. Please check all occurrences of cross-referencing carefully.
**We kept Sect. 2.1.2. because in this section it is mentioned the "We use $\overline{\sigma\varphi_0}$ = 129 a$^{-1}$ and μ = 0.596 mm$^{-1}$ that were determined from two calibration rock surfaces of similar granitic lithology from the Mont Blanc massif, with no erosion and known exposure age (Fig. A2). The values $\dot{D}$ = 8 Gy ka$^{-1}$ and $D_0$ = 500 Gy were selected as they are comparable to the average values obtained for samples used in this study."**

p.12, line 6: 'samples used in the following of this study (Table 3).' Delete 'the following'. Also, Table 3 does not show the averages for D-dot or D-zero. Which table are you referring to?
**Changed to "The value $\dot{D}$= 8 x 10$^{-3}$ Gy a$^{-1}$ was selected as average value obtained for samples used in this study ($\dot{D}$ = 7.4 and 8.4 x 10$^{-3}$ Gy a$^{-1}$ in Table 2)."**

p.13, line 2-3: the erosion rates 10$^{-2}$ mm a$^{-1}$ and 1 mm a$^{-1}$ do not appear in Sect. 2.1.2 as stated.
**We thank the reviewer for pointing this mistake, the erosion rates were mentioned in Section 2.1.2.3. This has been corrected.**

p.14, line 8: 'Figs. 6a, b, c, d' should be 'Figs. 7a, b, c, d'
**We thank the reviewer for pointing this mistake, we changed as suggested.**

p.14, line 17: delete 'but constant for an infinite'
**Changed as suggested.**

p.15, line 3: 'valid' should be 'validate'
**Changed as suggested.**

p.15, line 7: check Sect number
**Changed to Sect. 2.1.2.3.**

p.15, line 7-8: 'this range being arbitrarily decided even so the upper boundary is set to be approximately' should be 'this range being arbitrarily decided with the upper boundary set to approximately'
**Changed as suggested.**

p.15, line 12: 'parameters' should be 'parameter'
**Changed as suggested.**

p.15, line 15: delete 'further the limit laying in'
**Changed as suggested.**

p.18, Fig 8c: show the precise location of sample MBTP6. This is important to explain the shielding value in Table 3.
**The picture was changed and a white arrow have been placed to give a better view of the location of sample MBTP6.**

p.20, Table 2 caption, line 8: 'in between' should be 'between'. This happens twice in the line
**Changed as suggested.**

p.20, line 13-16: how is it possible that the calculated $t_0$ exposure age uncertainties are smaller than the measured cosmogenic nuclide concentration uncertainties.
**We thank the reviewer to point out this mistake, the uncertainties mentioned were representing the 1-sigma associated uncertainty, we replaced those uncertainties with 1-sigma uncertainty associated to the production rate.**

p.20, line 18: 'Figure 8' should be 'Figure 9'
**Changed as suggested.**

p.20, line 30: 'reference profile is lying at 23.5 mm' should be 'reference profile is at 23.5 mm'
**Changed as suggested.**

p.21, line 9: 'lies in between' should be 'lies between'
**Changed as suggested.**

p.21, line 9: '$\dot\varepsilon = 1$ mm a$^{-1}$' should be '$\dot\varepsilon = 10$ mm a$^{-1}$'
**We thank the reviewer for pointing this mistake, we corrected this.**

p.21, line 10: 'Sect. 3.2' should be 'Sect. 3.3'
**Changed as suggested.**

p.21, line 15: '$\dot\varepsilon = 1$ mm a$^{-1}$' should be '$\dot\varepsilon = 10$ mm a$^{-1}$'
**We thank the reviewer for pointing this mistake, this was corrected.**

p.23, line 20: 'erosion rate about' should be 'erosion rate of about'
**Changed as suggested.**

p.23, line 22: '(Rades el al. 2018) have showed' should be '(Rades et al. 2018) showed'
**Changed as suggested.**

p.24, line 9: 'for too long duration' should be 'for long durations'
**Changed as suggested.**

p.24, line 12: 'time $t_s$ pair' should be 'time $t_s$ pairs'
**Changed as suggested.**

p.24, line 24: '($\dot{\varepsilon}$ = 4.3 m a$^{-1}$ during $t_s$ = 4 years)' should be '($\dot{\varepsilon}$ = 4.3 mm a$^{-1}$ during $t_s$ = 4 years)', i.e. millimetres, not metres
**Corrected.**

p.24, line 27: 'limit our method' should be 'limits of our nethod'
**Changed as suggested.**

p.24, line 29: 'Such high difference of erosion between two locations of the same vertical profile is unlikely'. I think this statement is not supported by your data. Considering the difference in sample shielding it appears that MBTP6 was collected from a steeper slope than MBTP1. Fig. 3 suggests that the rock face may have lost mass by spallation, which could explain the order of magnitude lower $^{10}$Be concentration. These types of issues should be explored more.
**We thank the reviewer to point this, the sentence was changed to:**
**"Such high difference of erosion between two locations of the same vertical profile could be explain by the local topographic and environmental conditions such as slope surface and snow cover and controlling the efficiency of frost-cracking."**

p.24, line 34: 'The assumption that surface at 2094....almost 50 ka latter than...' should be 'The assumption that a surface at 2094....almost 50ka longer than...'
**Changed as suggested.**

p.25, line 1: 'latter' should be 'later'
**Changed as suggested.**

p.26, line 5: 'the correction TCN dating of erosion' should be 'an erosion correction for TCN dating'
**Changed as suggested.**

p.26, line 9: 'gab' should be 'gap'
**Corrected.**

p.28, Fig. A2 caption: 'These samples were in 2016 ...profiles' should be 'These samples were...profiles in 2016'
**Changed as suggested.**

p.28, Fig. A3 caption: 'comparable to the average values obtained...' What does comparable mean? What were the average values? Quantify "comparable".
**We replaced the sentence in the caption by "using […] similar values than Sohbati et al. 2018" to avoid confusion.**

p.29, Fig. A4 caption: 'exposure age obtains using' should be 'exposure age calculated using'
**Changed as suggested.**